



# Catchment attributes and meteorology for large sample study in contiguous China

Zhen Hao[2], Jin Jin[1,2], Runliang Xia[2], Shimin Tian[2], Wushuang Yang[2], Qixing Liu[2], Min Zhu[2], Tao Ma[2], Chengran Jing[2]

[1]School of Computer Science, Northwestern Polytechnical University, Xi'an, China, 710072
[2]Yellow River Institute of Hydraulic Research, Zhengzhou, China, 450003

*Correspondence to*: Jin Jin (jinjinhao@21cn.com)

**Abstract.**

We introduce the first large-scale catchment attributes and meteorological time series dataset of contiguous China. To develop
the dataset, we compiled diverse data sources to generate basin-oriented features describing the catchment characteristics related to hydrological processes. The proposed dataset consists of catchment characteristics, including soil, land cover, climate, topography, geology, and 29-year meteorological time series (from 1990 to 2018). The meteorological variables include precipitation, temperature, evapotranspiration, wind speed, ground surface temperature, pressure, humidity and sunshine duration. We also derived a daily potential evapotranspiration time series based on a modified Penman's equation. The studied
catchments are 4875 catchments within contiguous China derived from digital elevation models. We analysed and organised the spatial variations of catchment characteristics into a series of maps. Correlation analysis between attributes was conducted. Compared to the previously proposed datasets, we derived more catchment characteristics resulting in 125 attributes, providing a complete description of the catchments. Besides, we propose Normal-Camels-YR, a hydrological dataset covering 102 basins of the Yellow River basin with normalized streamflow observations. The proposed dataset provides numerous opportunities
for comparative hydrological research, such as examining the difference in hydrological behaviours across different catchments and building general rainfall-runoff modelling frameworks for many catchments instead of limited to a few. The dataset is freely available via http://doi.org/10.5281/zenodo.4704017 for community use. We will open-source the complement code for generating the dataset such that the user can generate meteorological series and catchment attributes for any watershed within contiguous China.

**1 Introduction**

Studying a large set of catchments often provides insights that cannot be obtained when looking at a single or few catchments (Coron, Andreassian et al. 2012, Kollat, Reed et al. 2012, Newman, Clark et al. 2015, Lane, Coxon et al. 2019). The hydrologic cycle consists of many sub-processes, including evaporation from the ocean, raindrop, interception, surface runoff, infiltration, etc. Catchment attributes such as soil characteristics, land cover characteristics and climate indices influence the water
movement and storage in these sub-processes such that hydrologic behaviours can vary across catchments (van Werkhoven,



Wagener et al. 2008). The same hydrological model may not be applicable in another basin. However, by examining a large sample of catchments, it is possible for the hydrological model to learn the similarities and differences of hydrological behaviours across catchments. For example, prediction in ungauged basins is a challenging problem present in hydrology. The central challenge is how to extrapolate hydrologic information from gauged basins to ungauged ones. Solving the problem

relies on understanding the similarities and differences between different catchments. However, regionally and temporally imbalanced observations bring a difficulty to the problem. For a hydrologic model to successfully simulate the ungauged areas, it must adapt itself to the different hydrologic behaviours present in different catchments. (Kratzert, Klotz et al. 2019) shows encoding catchment characteristics (e.g., soil characteristics, land cover, topography) into a data-driven model can teach model to behave differently responding the meteorological time series input based on different sets of static catchment attributes.


(Silberstein 2006, Shen, Laloy et al. 2018, Nevo, Anisimov et al. 2019) pointed out that large sample hydrological datasets are the foundation and key of many hydrological studies. The term big hydrologic data refers to all data influencing the water cycle, such as the meteorological variables, infiltration characteristics of the study area, land use or land cover types, physical and geological features of the study area, etc. Many studies cannot be carried out without large-scale hydrologic data (Coron,

Andreassian et al. 2012, Singh, van Werkhoven et al. 2014, Berghuijs, Aalbers et al. 2017, Gudmundsson, Leonard et al. 2019, Tyralis, Papacharalampous et al. 2019). For hydrological research, basin-orientated large sample datasets are of great significance. For example, comparative hydrology (de Araújo and González Piedra 2009, Singh, Archfield et al. 2014) focus on understanding how hydrological processes interact with the ecosystem, in particular, how hydrologic behaviours change under changes in the surface and sub-surface of the earth to determine to what extent hydrological predictions can be transferred

from one area to another. Large-sample catchment attributes dataset provide opportunities for research studying interrelationships among catchment attributes. (Seybold, Rothman et al. 2017) studied the correlations between river junction angle with geometric factors, downstream concavity, and aridity. (Oudin, Andréassian et al. 2008) investigates the link between land cover and mean annual streamflow based on 1508 basins representing a large hydroclimatic variety. (Voepel, Ruddell et al. 2011) examines how the interaction of climate and topography influences vegetation response.


Data-driven methods can best benefit from large-scale data. Data-driven approaches have shown great potential in various fields, transforming the applications in many industries (LeCun, Bengio et al. 2015). However, data-driven methods, especially the deep learning-based approaches, usually require high data volumes. Limited data will cause the over-fitting (Blumer, Ehrenfeucht et al. 1987, Abu-Mostafa, Magdon-Ismail et al. 2012) problem. Therefore, big hydrologic data is the fundamental

support for the successful deployment of powerful data-driven strategies.

Traditional hydrological models have some long standing challenges, such as the inability to capture hydrological processes' mechanism complexity (Kollat, Reed et al. 2012), which is due to the structural limitations of the conceptual models. Data-driven methods are proposed to overcome some existing obstacles. Data-driven strategies open a new way for researchers to



acquire knowledge transforming the research pattern from hypothesis-driven to data-driven. (Feng, Fang et al. 2020) proposed a flexible data integration fusing various types of observations to improve rainfall-runoff modelling. The research shows that combining different resources of data benefits predictions in regions with high autocorrelation in streamflow. (Wongso, Nateghi et al. 2020) developed a model predicting the state-level, per capita water uses in the United States, taking various geographic, climatic, and socioeconomic variables as input. The research also identified key factors associated with high water

usage. (Mei, Maggioni et al. 2020) proposed a statistical framework for spatial downscaling to obtain hyper-resolution precipitation data. The results show improvements compared with the original product. (Brodeur, Herman et al. 2020) applied machine learning techniques, namely bootstrap aggregation and cross-validation, to reduce overfitting in reservoir control policy search. (Ni and Benson 2020) proposed an unsupervised machine learning method to differentiate flow regimes and identify capillary heterogeneity trapping, showing the promise of machine learning methods for analysing large datasets from

coreflooding experiments. (Legasa and Gutiérrez 2020) propose to apply Bayesian Network for multisite precipitation occurrence generation. The proposed methodology shows improvements for existing methods.

World-wide data sharing has become a trend (Wickel, Lehner et al. 2007, Ceola, Arheimer et al. 2015, Blume, van Meerveld et al. 2018, Wang, Chen et al. 2020), and the amounts of hydrologic data available are ever-increasing. However, these data

typically came from different providers and are compiled in various formats. For example, ASTGTM[1] provides a global digital elevation model; GliM (Hartmann and Moosdorf 2012) includes rock types data globally; MODIS provides data products (Knyazikhin 1999, Didan 2015, Myneni, Knyazikhin et al. 2015, Running, Mu et al. 2017, Sulla-Menashe and Friedl 2018) describing features of the land and the atmosphere derived from remote sensing observations; (Yamazaki, Ikeshima et al. 2019) provides a global flow direction map at three arc-second resolution; HydroBASINS (Lehner 2014) provides basin boundaries

at different scales globally; and GDBD (Masutomi, Inui et al. 2009) provides basin boundaries with geographic attributes; GLHYMPS (Gleeson, Moosdorf et al. 2014) provides a global map of subsurface permeability and porosity; SoilGrids250m (Hengl, Mendes de Jesus et al. 2017) dataset provides global numeric soil properties. Local government agencies often hold meteorological data such as precipitation and evaporation, and the amount of this data is also growing, however, data transparency has still been a problem (Viglione, Borga et al. 2010). The data mentioned above are rarely spatially aggregated

to the catchment-scale, making it difficult for researchers to use these data. Properly pre-processed and formatted datasets on a large scale are of great importance for the hydrology research. Searching for appropriate data sources, pre-processing, and formatting often consumes a lot of researchers' time. In some cases, individual research groups either do not know where to obtain the appropriate data or cannot properly process the data to receive the desired format.

In summary, both data-driven and traditional hydrological research need diverse hydrologic datasets to learn the generalisation capability from one area to another. For a model to adapt to various behaviours in different catchments, the dataset must be

---

[1] https://asterweb.jpl.nasa.gov/gdem.asp



large enough to represent the complex heterogeneity presented in the natural hydrologic system. Although data sharing is being advocated in the community, it is usually difficult for the public to obtain certain data such as meteorological data and streamflow observations, either because there are not enough observations or because there are no open access permissions.


Recently, there are efforts (Addor, Newman et al. 2017, Alvarez-Garreton, Mendoza et al. 2018, Chagas, Chaffe et al. 2020, Coxon, Addor et al. 2020) compiling different types of data sources to form large scale hydrological datasets. These four collected datasets cover the continental United States, Chile, Brazil, and Great Britain. (Addor, Do et al. 2020) reviewed these datasets and discussed the guidelines for producing large-sample hydrological datasets and the limitations of the currently

proposed datasets. The CAMELS dataset has been used to support a lot of research. Based on CAMELS, (Kratzert, Klotz et al. 2018) built a Long Short-Term Memory (LSTM) network for rainfall-runoff modelling, showing that one model can predict the discharge for a variety of catchments. (Knoben, Freer et al. 2019) compared metrics used in hydrology based on simulations on many basins. (Tyralis, Papacharalampous et al. 2019) studied the relationship between the shape parameter and basin attributes based on the sizeable basin-oriented dataset.


However, there is no large-scale compilation of hydrological datasets in contiguous China. An alternative is on a global scale, the HydroATLAS (Linke, Lehner et al. 2019) dataset. However, since it is on a world-wide scale, compared with other datasets constructed for regions, the dataset lacks many attributes and is not built according to the CAMELS standards. Besides, the climatic data is not up to date (1950-2000), and the derivation of climatic data lacks ground surface observations inputs, such

that the data quality is not guaranteed.

Therefore, researchers still need to do repetitive works to compile data from different sources such as obtaining historical meteorological data (temperature, rainfall, evapotranspiration) of a catchment in contiguous China. Inspired by (Addor, Newman et al. 2017), in this paper, we present a catchment scale hydrologic dataset compiling a wide variety of hydrological

data, including basin topography, climate indices, land cover characteristics, soil characteristics and geological characteristics covering contiguous China.

The proposed dataset is the first dataset providing catchments meteorological time series and catchments attributes of contiguous China. We compiled and named the dataset following most standards of the previously proposed datasets. Unlike

CAMELS and CAMELS-CL, catchments in the proposed dataset are not selective. Instead, the dataset consists of all generated basins from the Digital Elevation Model (DEM), based on the Global Drainage Basin Dataset (Masutomi, Inui et al. 2009). The GDBD is derived at high-resolution (100m-1km) and has a good geographic agreement with existing global drainage





basin data in China[2]. Besides, an essential feature of the proposed dataset is that it provides a complete description of the catchment, rather than an abstraction. For example, both CAMELS and CAMELS-CL only report the most frequent and second

most frequent catchment land cover and lithology types. Instead, the proposed dataset calculates the proportion of each land cover and lithology type for each catchment to serve data-driven research better. We also introduced many more climate characteristics and soil characteristics to support more diverse potential research.

Researchers from different places can use the proposed dataset in conjunction with their streamflow data, simplifying

organising and compiling various data resources, which is usually repetitive work. The proposed dataset is undoubtedly the most comprehensive catchment attributes and meteorological time series dataset in contiguous China and is suitable for multi-purpose data-driven research. The dataset consists of basin boundaries in the shapefile format, computed catchment attributes of climate, land cover, soil, topography and lithology and 29-year meteorological time series. Table 1 compares the number of static attributes between CAMELS, CAMELS-BR, and the proposed dataset.


The paper is organized as follows: Section 2 describes the study area. Section 3-7 describes the five classes of the computed catchment attributes. In section 3-7, each unit follows the same structure: first introduce the meaning and significance of each added feature and data source used, then describe the variables' spatial variability if necessary. Section 8 describes the proposed catchment-scale meteorological forcing time series. Section 9 introduce the Normal-Camels-YR dataset, which provides

normalized streamflow measurements for 102 catchments of Yellow River. Section 10 describes the code and data availability. Section 11 presents the concluding remark.

In summary, our contributions are as follows:

(1) The proposed dataset is the first large-scale dataset containing catchment-scale meteorological time series of contiguous

China, which is the basis for many hydrological studies.

(2) We present the first basin-oriented static attributes dataset in contiguous China.

(3) We introduce several new catchment characteristics providing a complete description of the catchment compared with the previously proposed datasets such that the proposed dataset is prepared for potential hydrological studies.

(4) We offer a self-contained dataset covering 102 basins of the Yellow River basin with normalized runoff observation

supporting many potential studies.

---

[2] In this study, gauge streamflow measurements are not available in areas other than the Yellow River such that it is infeasible to specify a gauge location for generating the basin boundary for most of the areas. Streamflow measurements have strict redistribution policy; however, local research institutions have their streamflow measurements for hydrological research, the proposed dataset can used in conjunction with the streamflow data of researchers in various places.





(5) We will open-source the code for generating the dataset such that the user can generate a dataset for any watershed within contiguous China.

**Table 1 Number of computed attributes in CAMELS, CAMELS-BR and the proposed dataset.**

| Attribute class | CAMELS(A17) | CAMELS-BR | Ours |
|---|---|---|---|
| Location and topography | 9 | 11 | 12 |
| Geology | 7 | 7 | 18 |
| Soil | 11 | 6 | 54 |
| Land cover | 8 | 11 | 22 |
| Climatic indices | 11 | 13 | 17 |
| Human intervention indices | not computed | 4 | 2 |
| Total | 46 | 52 | 125 |

**Table 2 Summary of basin daily discharge and forcing data in CAMELS, CAMELS-BR and the proposed dataset.**

| Forcing data class | CAMELS | CAMELS-BR | Ours |
|---|---|---|---|
| Temperature | available | available | available |
| Precipitation | available | available | available |
| Solar radiation | available | **not available** | available |
| Day length | available | **not available** | **not available** |
| Sunshine hours | **not available** | **not available** | available |
| Humidity | available | **not available** | available |
| Snow water equivalent | available | **not available** | **not available** |
| Wind velocity | **not available** | **not available** | available |
| Ground surface pressure | available | **not available** | available |
| Observed evaporation | **not available** | available | available |
| Potential evapotranspiration | **not available** | available | available |
| Streamflow | available | available | partially available (see Section 9) |

## 2 Study area

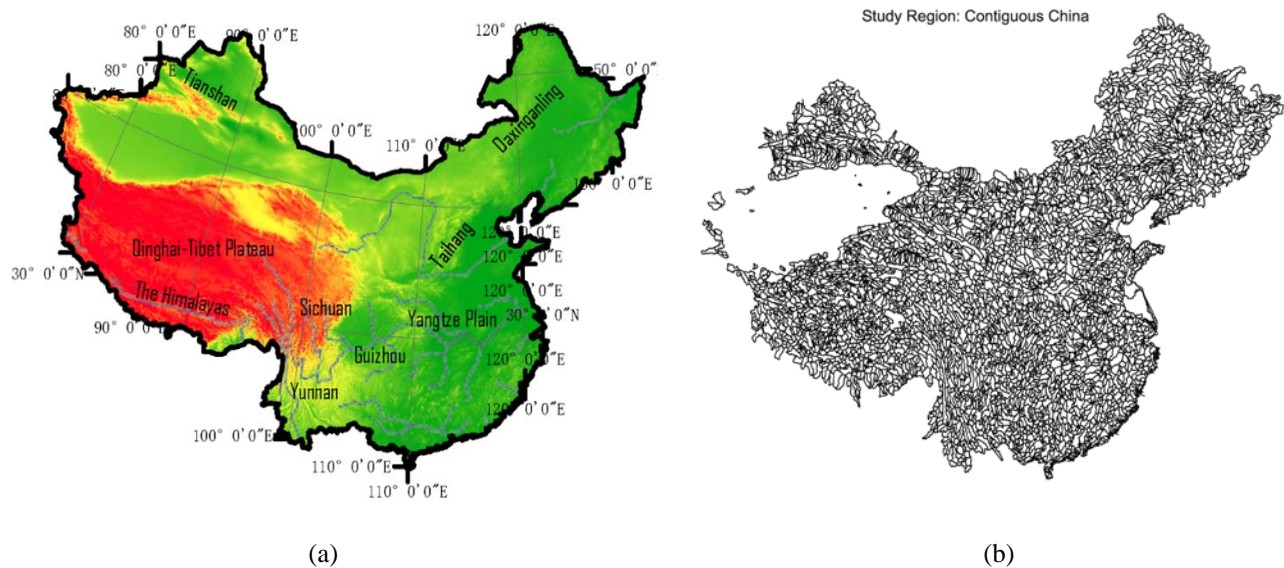

(a)                                              (b)

**Figure 1. Overview of the study area. The study area covers a wide range of latitude and longitude, from 18.2° N to 52.3° N, and from 76.0° E to 134.3° E. (a) The main geographical features map of contiguous China. China is mountainous; mountains and hills 165 occupy two-thirds of the area. (b) The distribution map of the delimited catchments based on the ASTER DEM, the catchments studied are all catchment areas delimited from the DEM, covering contiguous China, with 4875 catchments, most of which are 2000 to 5000 square kilometres.**

The study area corresponds to contiguous China, with diverse climate and terrain characteristics, spanning from 18.2° N to 52.3° N and 76.0° E to 134.3° E.  Mountains, plateaus, and hills account for about two-thirds of areas of contiguous China,

and the remaining are basins and plains. China's topography is like a three-level ladder, high in the west and low in the east. The Qinghai-Tibet Plateau, the highest plateau globally, located in the west of contiguous China, with a mean elevation of over 4000 meters, is the first step of China's topography. The Xinjiang region, the Loess Plateau, the Sichuan Basin, and the Yunnan-Guizhou Plateau to the north and east are the second step of China's topography. The mean sea level here is between 1000 to 2000 meters. Plains and hills dominate the east of the Daxinganling-Taihang Mountain to the coastline, the third step

of contiguous China. The elevation of this step descends to 500-1,000 meters.

In contiguous China, precipitation and temperature vary significantly in different places, forming a diverse climate environment. According to the Köppen Climate Classification System, from northwest to southeast, China's climate gradually evolves from Cold desert ($BW_k$) climate, Tundra (ET) climate, Warm and temperate continental ($D_{fa}$ and $D_{wb}$) climate to

Humid subtropical ($C_{wa}$) climate and Warm oceanic ($C_{fa}$) climate. From the perspective of temperature zones, there are tropical, subtropical, warm temperate, medium temperate, cold temperate and Qinghai-Tibet Plateau regions, and there are humid regions, semi-humid regions, semiarid regions, and arid regions from the perspective of wet and dry zones. Moreover, the same temperature zone can contain different dry and wet zones. Therefore, there will be differences in heat and wetness in the



same climate type. The complexity of the terrain makes the climate even more complex and diverse. Besides, China has a wide
range of regions affected by the alternating winter and summer monsoons. Compared with other parts of the world at the same
latitude, these areas have low winter temperatures, high summer temperatures, significant annual temperature differences, and
concentrated precipitation in summer. The cold and dry winter monsoon occurs in Asia's interior, far away from the ocean.
Under its influence, winter rainfall in most parts of China is low, accompanied by low temperature. The summer monsoon is
warm and humid, coming from the Pacific Ocean and the Indian Ocean. Under its influence, precipitation generally increases.

**Table 3 Summary table of catchment attributes available in the proposed dataset**

| Attribute class | Attribute name | Description | Unit | Data source |
|---|---|---|---|---|
| Climate indices (computed for 1 Oct 1990 to 30 Sep 2018) | pet_mean | mean daily pet (Penman–Monteith equation) | mm d$^{-1}$ | (Subramanya 2013) |
| | evp_mean | mean daily evaporation (observations) | mm d$^{-1}$ | SURF_CLI_CHN_MUL_DAY[3] |
| | gst_mean | mean daily ground surface temperature | °C | |
| | pre_mean | mean daily precipitation | mm d$^{-1}$ | |
| | prs_mean | mean daily ground surface pressure | hPa | |
| | rhu_mean | mean daily relative humidity | - | |
| | ssd_mean | mean daily sunshine duration | h | |
| | tem_mean | mean daily temperature | °C | |
| | win_mean | mean daily wind speed | m s$^{-1}$ | |
| | p_seasonality | seasonality and timing of precipitation (estimated using sine curves to represent the annual temperature and precipitation cycles, positive [negative] values indicate that precipitation peaks in summer [winter], values close to 0 indicate uniform precipitation throughout the year) | - | |

---

[3] http://data.cma.cn/data/cdcdetail/dataCode/SURF_CLI_CHN_MUL_DAY.html



| | | | | |
|---|---|---|---|---|
| | high_prec_freq | frequency of high-precipitation days ( $\geq$ 5 times mean daily precipitation) | d yr$^{-1}$ | |
| | high_prec_dur | average duration of high-precipitation events (number of consecutive days $\geq$ 5 times mean daily precipitation) | d | |
| | high_prec_timing | season during which most high-precipitation days ($\geq$ 5 times mean daily precipitation) occur | season | |
| | low_prec_freq | frequency of dry days (< 1mm d$^{-1}$) | d yr$^{-1}$ | |
| | low_prec_dur | average duration of dry periods (number of consecutive days < 1 mm d$^{-1}$) | d | |
| | low_prec_timing | season during which most dry days (< 1 mm d$^{-1}$) occur | season | |
| | frac_snow_daily | fraction of precipitation falling as snow (for days colder than 0 °C) | - | |
| | p_seasonality | seasonality and timing of precipitation, positive [negative] values indicate that precipitation peaks in summer [winter], values close to 0 indicate uniform precipitation throughout the year | - | |
| Geological characteristics | geol_porosity | subsurface porosity | - | (Gleeson, Moosdorf et al. 2014) |
| | geol_permeability | subsurface permeability (log-10) | m$^2$ | |
| | ig | fraction of the catchment area associated with ice and glaciers | - | (Hartmann and Moosdorf 2012) |
| | pa | fraction of the catchment area associated with acid plutonic rocks | - | |
| | sc | fraction of the catchment area associated with carbonate sedimentary rocks | - | |





| | | |
|---|---|---|
| su | fraction of the catchment area associated with unconsolidated sediments | - |
| sm | fraction of the catchment area associated with mixed sedimentary rocks | - |
| vi | fraction of the catchment area associated with intermediate volcanic rocks | - |
| mt | fraction of the catchment area associated with metamorphic | - |
| ss | fraction of the catchment area associated with siliciclastic sedimentary rocks | - |
| pi | fraction of the catchment area associated with intermediate plutonic rocks | - |
| va | fraction of the catchment area associated with acid volcanic rocks | - |
| wb | fraction of the catchment area associated with water bodies | - |
| pb | fraction of the catchment area associated with basic plutonic rocks | - |
| vb | fraction of the catchment area associated with basic volcanic rocks | - |
| nd | fraction of the catchment area associated with no data | - |
| py | fraction of the catchment area associated with pyroclastic | - |
| ev | fraction of the catchment area associated with evaporites | - |



| | | | | |
|---|---|---|---|---|
| Land cover characteristics | lai_max | maximum monthly mean of the leaf area index (based on 12 monthly means) | - | (Myneni, Knyazikhin et al. 2015) |
| | lai_diff | difference between the maximum and minimum monthly mean of the leaf area index (based on 12 monthly means) | - | |
| | ndvi_mean | mean normalized difference vegetation index (NDVI) | - | (Didan 2015) |
| | root_depth_50 | root depth (percentiles=50% extracted from a root depth distribution based on IGBP land cover) | m | Eq. 2 and Table 2 in (Zeng 2001) |
| | root_depth_99 | root depth (percentiles=99% extracted from a root depth distribution based on IGBP land cover) | m | |
| | evergreen needleleaf tree | catchment area fraction covered by evergreen needleleaf tree | - | (Sulla-Menashe and Friedl 2018) |
| | evergreen broadleaf tree | catchment area fraction covered by evergreen broadleaf tree | - | |
| | deciduous needleleaf tree | catchment area fraction covered by deciduous needleleaf forests | - | |
| | deciduous broadleaf tree | catchment area fraction covered by deciduous broadleaf tree | - | |
| | mixed forest | catchment area fraction covered by mixed forest | - | |
| | closed shrubland | catchment area fraction covered by closed shrubland | - | |
| | open shrubland | catchment area fraction covered by open shrubland | - | |
| | woody savanna | catchment area fraction covered by woody savanna | - | |



| | savanna | catchment area fraction covered by savanna | - | |
|---|---|---|---|---|
| | grassland | catchment area fraction covered by grassland | - | |
| | permanent wetland | catchment area fraction covered by permanent wetland | - | |
| | cropland | catchment area fraction covered by cropland | - | |
| | urban and built-up land | catchment area fraction covered by urban and built-up land | - | |
| | cropland/natural vegetation | catchment area fraction covered by cropland/natural vegetation | - | |
| | snow and ice | catchment area fraction covered by snow and ice | - | |
| | barren | catchment area fraction covered by barren | - | |
| | water bodies | catchment area fraction covered by water bodies | - | |
| Topography, location, and Human intervention | basin_id | drainage basin identifiers | - | (Masutomi, Inui et al. 2009) |
| | pop | population | people | |
| | pop_dnsty | population density | people km$^{-2}$ | |
| | lat | mean latitude | °N | |
| | lon | mean longitude | °E | |
| | elev | mean elevation | M | |
| | area | catchment area | km$^2$ | |
| | slope | mean slope | m km$^{-1}$ | (Horn 1981) |
| | length | The length of the mainstream measured from the basin outlet to the remotest point on the basin boundary. The mainstream is identified by starting from the basin | km | (Subramanya 2013) |





| | | | | |
|---|---|---|---|---|
| | | outlet and moving up the catchment. | | |
| | form factor | catchment area / (catchment length)$^2$ | - | |
| | shape factor | (catchment length)$^2$ / catchment area | - | |
| | compactness coefficient | perimeter of the catchment / perimeter of the circle whose area is that of the basin | - | |
| | circulatory ratio | catchment area / area of circle of catchment perimeter | - | |
| | elongation ratio | diameter of circle whose area is basin area / catchment length | - | |
| Soil | pdep | soil profile depth | cm | (Shangguan, Dai et al. 2013) |
| | clay | percentage of clay content of the soil material | % | |
| | sand | percentage of sand content of the soil material | % | |
| | por | porosity | cm$^3$ cm$^{-3}$ | |
| | silt | percentage of silt content of the soil material | % | |
| | grav | rock fragment content | % | |
| | som | soil organic carbon content | % | |
| | log_k_s[4] | log-10 transformation of saturated hydraulic conductivity | cm d$^{-1}$ | (Dai, Xin et al. 2019) |
| | theta_s[4] | saturated water content | cm$^3$ cm$^{-3}$ | |
| | tksatu[4] | thermal conductivity of unfrozen saturated soils | W m$^{-1}$ K$^{-1}$ | |
| | bldfie[4] | bulk density | kg m$^{-3}$ | |

---

[4] The data source contains multi-layer soil data, soil characteristics for all layers are determined.





| cecsol[4] | cation-exchange capacity | cmol+ kg⁻¹ | (Hengl, Mendes de Jesus et al. 2017) |
|---|---|---|---|
| orcdrc[4] | organic carbon content | g kg⁻¹ | |
| phihox[4] | pH in H2O | 10⁻¹ | |
| bdticm | depth to bedrock | cm | |

# 3 Climate indices

Meteorological raw data was provided by the China Meteorological Data Network[3], released as the SURF_CLI_CHN_MUL_DAY (V3.0) dataset, which provides complete variable types and the longest period (1951-2018) of
meteorological time series of China. The SURF_CLI_CHN_MUL_DAY product includes site observations of pressure, temperature, relative humidity, precipitation, evaporation, wind speed, sunshine duration, and ground surface temperature. The summary is presented in Table 4. The Inverse distance weighting method is used for interpolating the site observations. Climate indices are then obtained by taking the average of the catchment-scale extraction from the interpolated raster. To ensure data quality, we chose the latter 29-year record (from 1990 to 2018) to construct the dataset since sites' distribution was sparse in
the early days (Fig. 2). We computed more climatic characteristics compared with other datasets (Table 2). These characteristics have critical potential effects on the hydrological processes; for example, wind speed can affect actual evapotranspiration. To be consistent with the CAMELS (Addor, Newman et al. 2017), we also determined all climatic attributes (Woods 2009) in the CAMELS dataset. The proposed dataset provides more meteorological variables and longer time series (1990-2018) than CAMELS and CAMELS-CL. A summary of the computed Climate indices is presented in Table
3. The national distribution of meteorological attributes of catchments is shown in Fig. 3.

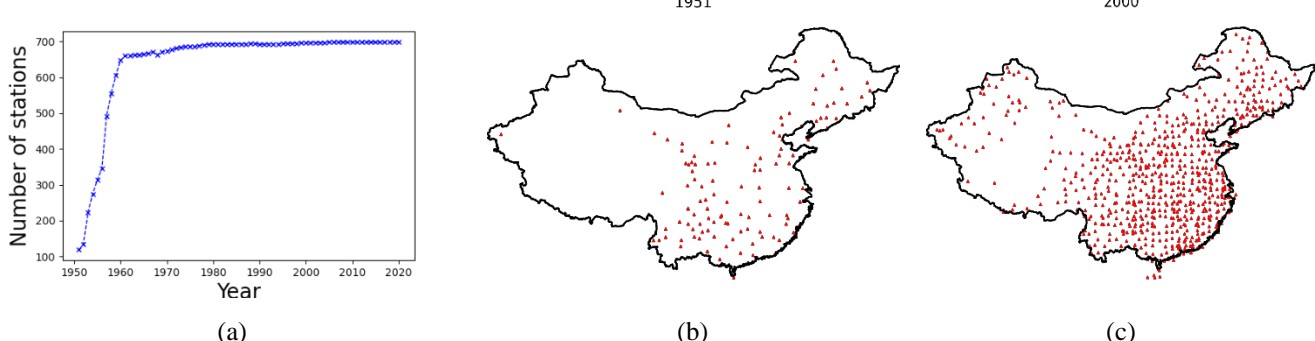

Figure 2. Overview of changes in the number and distribution of meteorological stations in China. (a) The number of meteorological stations varies with the year. There were only 119 stations in 1951. This number increased rapidly from 1951 to the early 1960s, and the number of stations remained stable after 2000. (b) Distribution map of China's meteorological stations in 1951. (c) Distribution map of China's meteorological stations in 2000.





(a)

(b)

(c)

(d)

(e)

(f)

(g)

(h)

(i)

(j)

(k)

(l)



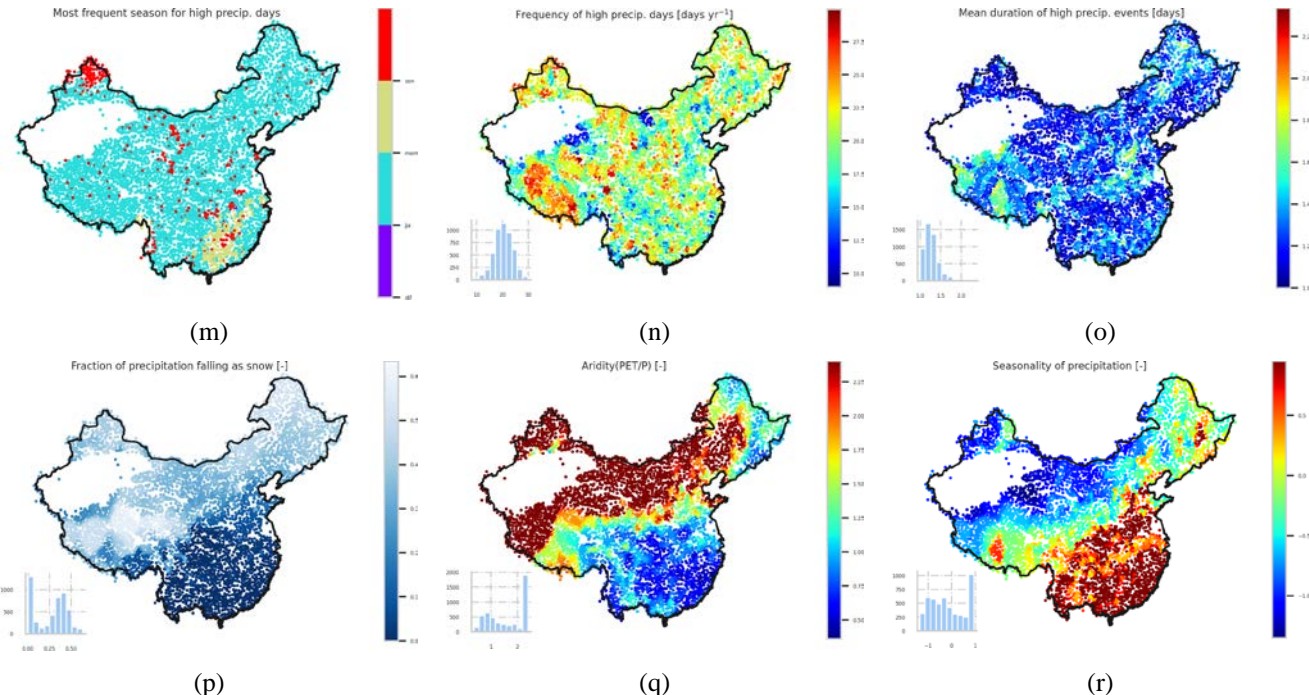

**Figure 3. Maps of climatic indices over contiguous China. The histograms and bar plots indicate the number of catchments (out of 4875) in each bin or category.**

The instruments for measuring evaporation were updated from 2000 to 2005. Early observations can be multiplied by a correction coefficient to approximate the new tools. However, the coefficient varies across stations making the approach infeasible. To complement this, we calculated potential evapotranspiration (PET) based on a modified Penman's Equation (see Appendix) and other observed meteorological variables, providing a series of consistent evapotranspiration estimation.

The average daily precipitation in contiguous China is highest in the southeast and lowest in the northwest. It is also higher in the coastal areas than in the interior land. Ground surface pressure is positively correlated with elevation, the highest in the Qinghai-Tibet Plateau and the lowest in the Southeast Plain. The average relative humidity is generally positively correlated with precipitation; they are also higher in some forested areas, such as the Taihang Mountains and Daxingan Mountains. The Qinghai-Tibet Plateau has the lowest average temperature, and the southern coastal area has the highest. A distinctive feature of the distribution of wind speed is the high wind speed in mountainous areas. The highest wind speed occurs in the southeast coastal area (> 6 meters per second). Refer to Section 8 for a detailed description of the proposed catchment-scale meteorological time series dataset of contiguous China.





## 4 Geology


To describe the lithological characteristics of each catchment, we used the same two global datasets as CAMELS, Global Lithological Map (GLiM) (Hartmann and Moosdorf 2012) and GLobal HYdrogeology MaPS (GLHYMPS) (Gleeson, Moosdorf et al. 2014). Figure 4 presents the results.


GLiM provides a high resolution global lithological map assembled from existing regional geological maps; it has been widely used for constructing datasets (e.g. SoilGrids250m (Hengl, Mendes de Jesus et al. 2017)). However, the data quality of GLiM can vary in different spatial locations depending on the quality of the original regional geological maps. GLiM consists of three levels, the first level contains 16 lithological classes, and the additional two levels describe more specific lithological characteristics. For contiguous China, the compiled regional data sources (China 1991, Xinjiang 1992, Survey 2001) have


slightly lower resolutions than the GLiM target resolution (1:1 000 000). However, for a basin-scale study with a mean basin area of over 2000 km$^2$, the classification accuracy should satisfy most applications.

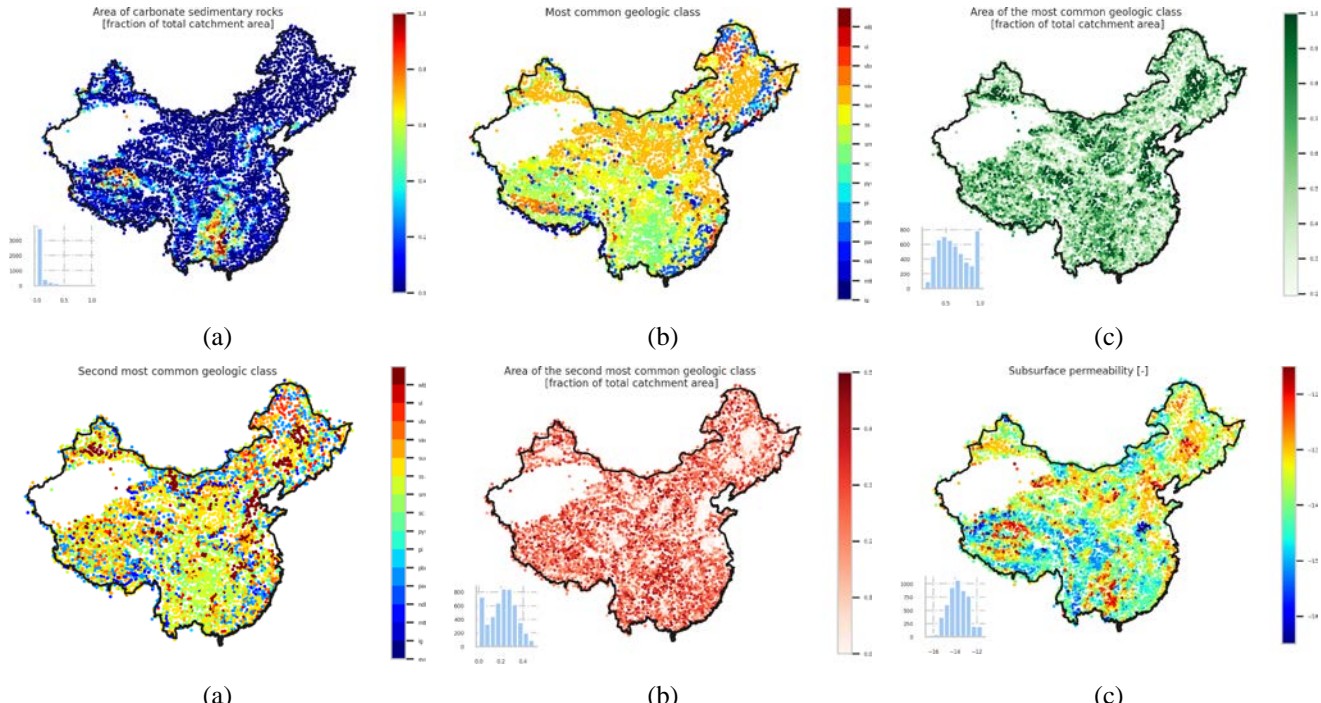

(a)                          (b)                          (c)

(a)                          (b)                          (c)

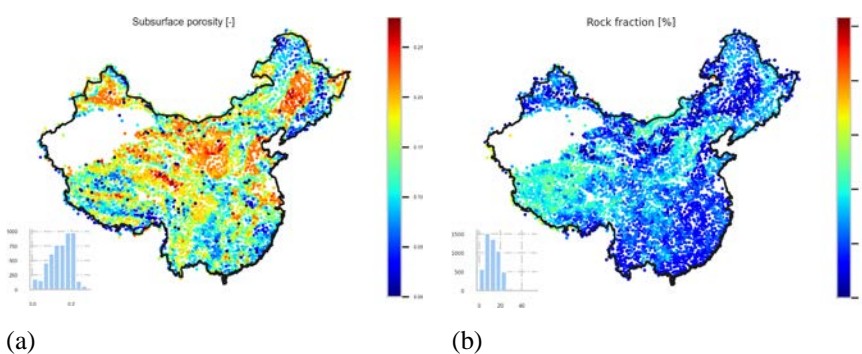

(a)                                            (b)

**Figure 4. Maps of geological characteristics over contiguous China. The histograms indicate the number of catchments (out of 4875) in each bin.**

Compared to CAMELS and CAMELS-CL, one design consideration of the proposed dataset is that it should be more prepared
for the data-driven research, such that we aim to generate as many types of catchment-scale data as possible since advanced
data-driven methods can learn the representation of inputs automatically. To this end, we determined and recorded each
lithological class's contribution to the catchment instead of recoding just the first and second most frequent classes. The GLiM
is represented by 1,235,400 polygons; the polygons are converted to raster format for the basin-scale lithological type statistics.

GLobal HYdrogeology MaPS (GLHYMPS) provides a global estimation of subsurface permeability and porosity, two critical
characteristics for the soils' hydrological classification. Porosity and permeability influence an area's infiltration capacity. Soil
with high porosity is likely to contain s amounts of water, and high permeable soil transmits water relatively quickly. Based
on the high-resolution map of GLiM, which can differentiate fine and coarse-grained sediments and sedimentary rocks,
GLHYMPS determined subsurface permeability depending on the different permeabilities of rock types. For the proposed
dataset, we calculated the catchment arithmetic mean for porosity. Followed (Gleeson, Smith et al. 2011), the logarithmic scale
geometric mean is used for representing subsurface permeability. The summary of geological characteristics is present in Table
3.

Porosity and permeability have similar distributions as geological classes. These two characteristics are highly dependent on
rock properties, unconsolidated sediments, mixed sedimentary rocks, siliciclastic sedimentary rocks, carbonate sedimentary
rocks, and acid plutonic rocks are the five most common geological classes in contiguous China. Unconsolidated sediment is
the most common rock type in contiguous China, dominating 31.9% of catchments; it extends from Xinjiang to the inland of
the northeast and the coastal area surrounding the Bohai Sea, due to the high proportion of unconsolidated sediments present
in the rock, these areas typically have high permeability and medium porosity. Mixed sedimentary rocks are the second most
common rock type in contiguous China, accounting for 20.3% of catchments, it dominated the southern Qinghai-Tibet Plateau,
western Yunnan-Guizhou Plateau, and northern Inner Mongolia. These areas typically have high porosity and low permeability.
Siliciclastic sedimentary rocks dominate 17.7% of basins, mainly distributed in the northern part of the Qinghai-Tibet Plateau





and the junction of the Qinghai-Tibet Plateau and the Yunnan-Guizhou Plateau; there are also some distributions in the eastern inland. These areas have low subsurface permeability and high subsurface porosity. Amongst all catchments, 9.8% of
catchments are dominated by carbonate sedimentary rocks. Carbonate sedimentary rocks are mainly located in eastern Yunnan and northern Qinghai-Tibet Plateau. Acid plutonic rocks are typically distributed in the mountains surrounding the inland northeast, namely the Daxinganling Mountain and the hills in southern Guangdong and southwestern Guangxi. They are also distributed along the Brahmiputra river in the south part of the Qinghai-Tibet Plateau. The distribution of Acid plutonic rocks is relatively scattered; there are many isolated Acid plutonic rocks distributions in different locations of contiguous China,
accompanied by medium permeability and high porosity.

In summary, the types of rocks in contiguous China are dominated by unconsolidated sediments and mixed sedimentary rocks. In 33.86% of the catchments, the dominant rock types occupy less than 50% of the catchment areas, and only 16.8% basins are having a dominant rock type with an area fraction greater than 90%. Amongst 4875 basins, 9.4% of basins have prevalent
rock types wholly occupying the area.

## 5 Landcover

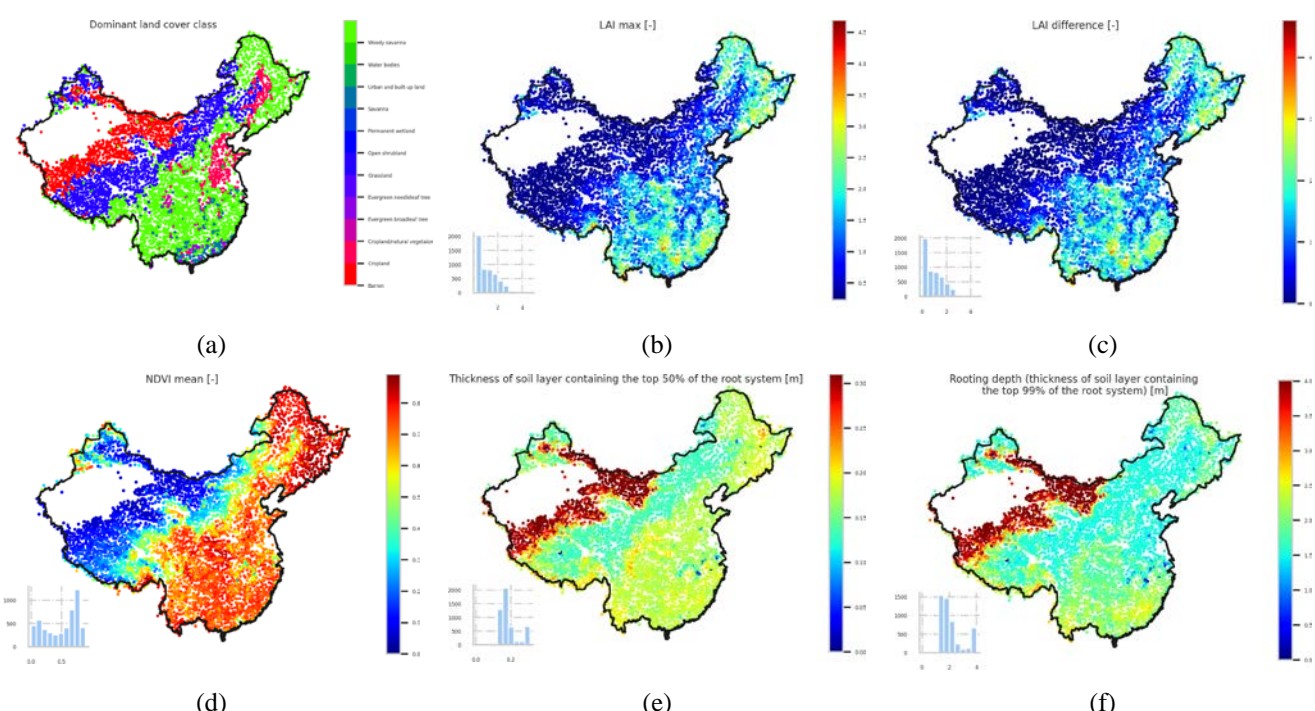

(a)                          (b)                          (c)

(d)                          (e)                          (f)





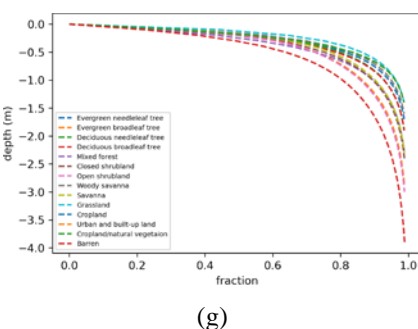

(g)

**Figure 5. Maps of land cover characteristics over contiguous China. The histograms indicate the number of catchments (out of 4875) in each bin.**

We selected two indicators to characterize vegetation density and growth on the surface: Normalized difference vegetation
index (NDVI) and Leaf area index (LAI). NDVI is an indicator with a valid range of -0.2 to 1, assessing whether the area being observed contains live green vegetation or the plants' health. However, NDVI is just a qualitative measurement of the vegetation density; it cannot provide a quantitative estimate of the vegetation density in the area. Moreover, NDVI often provides inaccurate vegetation density measurements, and only long-term measurement and comparison can ensure its accuracy. NDVI alone is not enough to estimate the state of plants in an area. Therefore, we have selected another indicator,
LAI, to supplement the deficiencies of NDVI.

LAI is defined as the total needle surface area per unit ground area and half of the entire needle surface area per unit ground surface area. It is a quantifiable value. It is functionally related to many hydrological processes like water interception (van Wijk and Williams 2005). (Buermann, Dong et al. 2001) verifies the validity of LAI used to characterize vegetation growth.
The data sources used are The Terra Moderate Resolution Imaging Spectroradiometer (MODIS) Vegetation Indices (Didan 2015) for NDVI and Moderate Resolution Imaging Spectroradiometer (MODIS) (Myneni, Knyazikhin et al. 2015) for LAI. Followed (Addor, Newman et al. 2017), we determined maximum monthly LAI as an indicator characterising vegetation interception capacity and the maximum evaporative capacity and the difference between the maximum and minimum monthly LAI representing LAI's temporal variations.


Land cover classification refers to segmenting the ground into different categories based on remote sensing images. The Terra and Aqua combined Moderate Resolution Imaging Spectroradiometer (MODIS) Land Cover Type provides different results depending on the classification system used. Annual International Geosphere-Biosphere Programme (IGBP) classification is used for building the dataset, which is derived by the c4.5 decision tree algorithm. The IGBP classification system was
formulated by the IGBP Land Cover Working Group in 1995, resulting in 17 categories of land cover types (Belward, Estes et al. 1999). (Friedl, Sulla-Menashe et al. 2010) compared the IGBP data of MODIS with other reference dataset and concluded that the MODIS classification of IGBP has an accuracy of 75%. We determined the fraction of each land cover class for each basin based on the Terra and Aqua combined Moderate Resolution Imaging Spectroradiometer (MODIS) Land Cover Type

(Sulla-Menashe and Friedl 2018), which differentiates our dataset from CAMELS and CAMELS-CL (only calculated the

proportion of the dominant types).

Followed (Addor, Newman et al. 2017), we also computed the average rooting depth (50% and 90%) for each catchment based

on the IGBP classification using a two-parameter method (Zeng 2001). The root depth distribution of vegetation affects the

ground's water holding capacity and the topsoil layer's annual evapotranspiration (Desborough 1997). Many models use root

depth as an essential parameter to characterize soil moisture absorption capacity. (Zeng 2001) developed a two-parameter

asymptotic equation for estimating root depth distribution; the root depth distribution is global, derived based on the IGBP

classification avoiding the problem of significantly different root distributions in various research. Figure 5(g) shows root

depth distributions of different vegetation types, based on (Zeng 2001)'s method. The 90% root depth is usually considered to

be "rooting depth", among the 17 categories of IGBP, cropland has the smallest rooting depth, and open shrubland has the

largest.  The 90% root depth of all vegetation is less than 2 meters. The national distribution of catchments soil characteristics

is shown in Fig. 5.

## 6 Location and topography

The catchments' boundary files are obtained from the global drainage basin dataset (Masutomi, Inui et al. 2009). The PDBD

dataset was derived from digital elevation models (DEMs) with a high-resolution (100m-1km), and the errors were corrected

by either automatic methods or manually. Additionally, PDBD also provides population and population density estimates for

catchments, and these two indicators are also included in our dataset as a measure of human intervention. Global Runoff Data

Centre (Center 2005) discharge gauging stations were used for referencing the derived basins. In contiguous China, PDBD has

a high average match area rate (AMAR) and good geographic agreement with existing global drainage basin data. Based on

the high-quality dataset, precise geographic and topographic information can be derived. See Fig. 6 for a summary.


The topography attributes of each catchment are determined based on the ASTGTM product retrieved from

https://lpdaac.usgs.gov, maintained by the NASA EOSDIS Land Processes Distributed Active Archive Center (LP DAAC) at

the USGS Earth Resources Observation and Science (EROS) Center.




(a)   (b)   (c)

(d)   (e)   (f)

(g)   (h)

**Figure 6. Maps of topographic characteristics over contiguous China. The histograms indicate the number of catchments (out of 4875) in each bin.**

The CAMELS dataset just provides two parameters (two area estimates) for describing the catchment shape; however, the physical characteristics of a catchment can affect the runoff volume and the runoff hydrograph of the catchment under a storm. To provide a complete description of the catchment shape, we computed several geometrical parameters of the catchment related to the runoff process, including catchment form factor, shape factor, compactness coefficient, circulatory ratio and the elongation ratio (Subramanya 2013). A summary of the location and topography attributes can be found in Table 3.





## 7 Soil

The proposed dataset has a total of 54 soil attributes (Table 3) derived from (Hengl, Mendes de Jesus et al. 2017), (Dai, Xin et al. 2019) and (Shangguan, Dai et al. 2013). The summary result is shown in Fig. 7. Five categories of soil characteristics (pH

in H2O, organic carbon content, depth to bedrock, cation-exchange capacity, and bulk density) are determined from SoilGrids. SoilGrids (Hengl, Mendes de Jesus et al. 2017) provides global predictions for soil properties including organic carbon, bulk density, cation exchange capacity (CEC), pH, soil texture fractions and coarse fragments by fusing multiple data sources including MODIS land products, SRTM DEM, climatic images and global landform and lithology maps at the 250m resolution. SoilGrids made predictions based on machine learning algorithms and many covariates layers primarily derived from remote

sensing data. SoilGrids has soil characteristics for several soil depths.

(a)                                    (b)                                    (c)

(d)                                    (e)                                    (f)

(g)                                    (h)                                    (i)



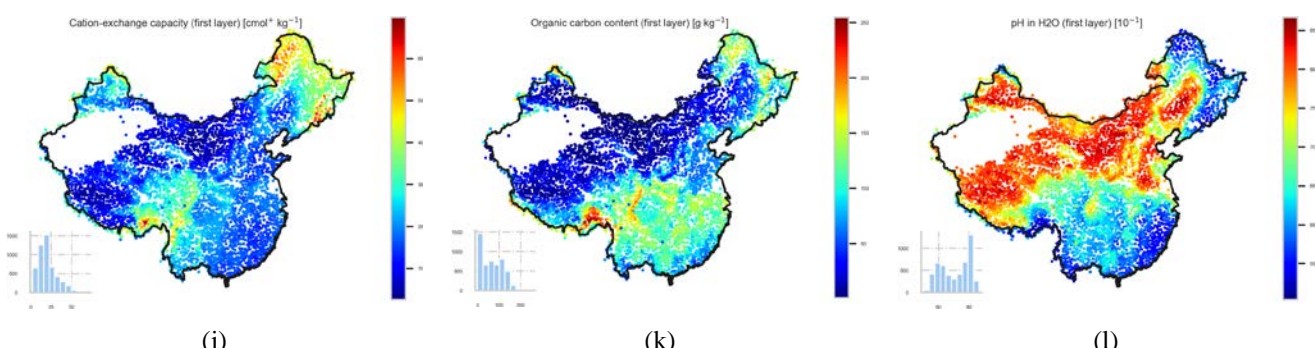

**Figure 7. Maps of soil characteristics over contiguous China. The histograms indicate the number of catchments (out of 4875) in each bin.**

Unlike CAMELS, whose reported results are obtained by a linear weighted combination of the different soil layers, and
CAMELS-BR, whose products are soil characteristics at a depth of 30cm. We computed soil characteristics at all soil layers provided by SoilGrids such that advanced models can learn directly from the raw inputs.

To be consistent with CAMELS, we also determined saturated water content and saturated hydraulic conductivity (Dai, Xin et al. 2019). We also introduced thermal conductivity of unfrozen saturated soils (Dai, Xin et al. 2019). (Dai, Xin et al. 2019)
provides a global estimation of soil hydraulic and thermal parameters using multiple Pedotransfer Functions (PTFs) based on SoilGrids. Based on the SoilGrids and GSDE (Shangguan, Dai et al. 2014) datasets, (Dai, Xin et al. 2019) produced six soil layers with a spatial resolution 30×30 arc-second. The vertical resolution of (Dai, Xin et al. 2019) is the same as the SoilGrids, with six intervals of 0–0.05 m, 0.05–0.15 m, 0.15–0.30 m, 0.30–0.60 m, 0.60–1.00 m, and 1.00–2.00 m. Same as the methods applied to SoilGrids, we determined and records catchment soil characteristics for all these layers.


To provide even more complete description of the soil, we determined seven more soil characteristics (Shangguan, Dai et al. 2013) including soil profile depth, porosity, clay/silt/sand content, rock fragment, and soil organic carbon content. (Shangguan, Dai et al. 2013) provides physical and chemical attributes of soils derived from 8979 soil profiles at 30×30 arc-second resolution, the polygon linkage method was used to derive the spatial distribution of soil properties. The profile attribute
database and soil map are linked under a framework avoiding uncertainty in taxon referencing.

Depth to bedrock controls many physical and chemical processes in soil. The distribution of depth to bedrock in contiguous China is characterised by (i) low in the mountainous areas, such as Yunnan province and Chongqing City; (ii) high in barren areas, e.g. North and Northwest China. The introduced soil pH value is crucial since it influences many other physical and
chemical soil characteristics. The spatial variability of soil pH in contiguous China is characterised by (i) soils in southern contiguous China are acid to strongly acid; (ii) soils in northern China are natural or alkaline; (iii) soils in north-eastern forested areas are also acid (pH < 7.2). Cation exchange capacity can be seen as a measure of soil fertility since it measures how much





nutrient the soil can store such that it influences the growth of the vegetations. Cation exchange capacity is positively correlated with soil organic matter content and clay content, which Cation exchange capacity is generally low in sandy and silty soils.

The spatial variability of Cation exchange capacity in contiguous China is characterised by (i) high in peat and forested areas in Qinghai-Tibet Plateau, central and northeast China (ii) The Cation exchange capacity in the desert area such as the northwest is extremely low. Soil hydraulic and thermal properties are greatly affected by soil organic matter (SOM). Soil organic matter has a similar distribution to the cation exchange capacity: high in the peat and forested areas such as northeast China and low in the north and northwest.

**8 Meteorological time series**

**Table 4 Summary table of catchment meteorological time series available in the proposed dataset**

| Variable | Description | Unit |
| --- | --- | --- |
| prs | catchment daily averaged ground pressure | hPa |
| tem | catchment daily averaged temperature at 2 m above ground | °C |
| rhu | catchment daily averaged relative humidity | - |
| pre | catchment daily averaged precipitation | mm d$^{-1}$ |
| evp | catchment daily averaged evaporation measured by ground instruments | mm d$^{-1}$ |
| win | catchment daily averaged wind speed at 2 m above ground | m s$^{-1}$ |
| ssd | catchment daily averaged sunshine duration | h d$^{-1}$ |
| gst | catchment daily averaged ground surface temperature | °C |
| pet | catchment daily averaged potential evapotranspiration determined by Penman's equation (see Appendix A) | mm d$^{-1}$ |

There have been many studies based on SURF_CLI_CHN_MUL_DAY in China (Liu, Xu et al. 2004, Xu, Gao et al. 2009, Huang, Han et al. 2016, Liu, Zheng et al. 2017), such as trend analysis of the pan evaporation (Liu, Yang et al. 2010). Still,

there has not yet been a large-scale basin-oriented meteorological time series dataset in contiguous China. Researchers still need to do repeated works to extract historical meteorological data from the SURF_CLI_CHN_MUL_DAY dataset for the research. For the first time, we release a catchment-scale meteorological time series dataset. We will also open-source the code for researchers to generate any catchment's meteorological time series within contiguous China. The basin-oriented dataset provides meteorological time series for 4875 basins from 1990 to 2018 based on the China Meteorological Data Network.

Meteorological time series includes pressure, temperature, relative humidity, precipitation, evaporation, wind speed, sunshine duration, ground surface temperature and potential evapotranspiration (see Table 4 for a summary).





The meteorological time series data from 1951 to 2010 is derived based on the "1951-2010 China National Ground Station Data Corrected Monthly Data File Basic Data Collection" data construction project. Other data include monthly reported data to the National Meteorological Information Centre by the provinces, and hourly and daily data uploaded by automatic ground stations in real-time. The SURF_CLI_CHN_MUL_DAY dataset is quality controlled, the quality and completeness of each variable are significantly improved compared to the previous similar products. M the development of the dataset, missing data were filled by interpolating its nearest stations.

Figure 2 presents the variation of the distribution of the observation sites. The start date of the recording is 1951, but because the early site distribution is sparse, we only used records from 1990 to 2018 to construct the dataset to ensure the data quality. The interpolation method used is the Inverse distance weighting since it shows better performance than other comparators. Catchment-scale raster is extracted from the interpolated national raster using the open-source rasterio[5] package. For all variables, we take the arithmetic mean on the extracted catchment raster as the catchment mean. Potential evapotranspiration (PET) is estimated based on Penman's Equation and other catchment meteorological variables.

## 9 Normal Camels YR – Normalized Catchment attributes and meteorology for Yellow River basin

Apart from the dataset providing the catchment attributes and meteorological forcing for contiguous China, we also offer a self-contained dataset covering the Yellow River basin with normalized streamflow measurements. The streamflow data are normalized to have zero mean and a standard deviation of 1 for each basin. The Normal-Camels-YR dataset is designed to support machine learning and deep learning research related to hydrology. In particular, fifty-four watersheds are less affected by human activities (selection is based on the Global Reservoirs and Dam databases (GRanD) (Lehner, Liermann et al. 2011) which provides the locations of reservoirs and dams globally), which makes them suitable for rainfall-runoff modelling research. For most machine learning and deep learning algorithms, data normalization will not affect model performance (e.g., neural network-based and tree-based algorithms). Besides, other research, such as trend analysis, can also be carried out. The Normal-Camels-YR dataset is self-contained to fully describe the Yellow River basin and is particularly helpful for the hydrology research of the Yellow River.

During the dataset development, basins with too few observations are removed, resulting in discontinuous basin identifiers. Normal-Camels-YR covers 102 gauges in the Yellow River basin, providing basin boundary shapefiles, static attributes and normalized streamflow measurements for each basin. The covered basins have areas ranging from 134 to 804,421 square kilometres. The time resolution of streamflow measurements is seven days, and the mean length of records of the streamflow measurements is 684, which means the mean period of the streamflow measurements for each basin is over 13 years.

---

[5] https://github.com/mapbox/rasterio





Meteorological variables included in Normal-Camels-YR is slightly different; it introduced daily maximum and minimum for some variables (Table 5).

425

**Table 5 Meteorological variables provided in Normal-Camels-YR, the time series length is 22 years (1999-2020)**

| Attribute name | Description | Unit |
| --- | --- | --- |
| evp | catchment daily averaged evaporation (observations) | 0.1 mm d$^{-1}$ |
| gst_mean | catchment daily averaged ground surface temperature | 0.1 °C |
| gst_min | catchment daily minimum ground surface temperature | 0.1 °C |
| gst_max | catchment daily maximum ground surface temperature | 0.1 °C |
| pre | catchment daily averaged precipitation | 0.1 mm d$^{-1}$ |
| prs_mean | catchment daily averaged ground surface pressure | 0.1 hPa |
| prs_max | catchment daily maximum ground surface pressure | 0.1 hPa |
| prs_min | catchment daily minimum ground surface pressure | 0.1 hPa |
| rhu | catchment daily averaged relative humidity | - |
| ssd | catchment daily averaged sunshine duration | 0.1 h |
| tem_mean | catchment daily averaged temperature | 0.1 °C |
| tem_min | catchment daily minimum temperature | 0.1 °C |
| tem_max | catchment daily maximum temperature | 0.1 °C |
| win_max | catchment daily maximum wind speed | 0.1 m s$^{-1}$ |
| win_mean | catchment daily averaged wind speed | 0.1 m s$^{-1}$ |

## 10 Data availability and software packages used.

The proposed dataset is freely available at http://doi.org/10.5281/zenodo.4704017. The files provided are (i) several separate

430 files containing 120+ catchments attributes, (ii) the daily meteorological time series in a zip file, (iii) the catchment boundaries used to compute the attributes and extract the time series, (iv) the Normal-Camels-YR dataset, (v) an attribute description file and (v) a readme file. The code used to generate the dataset is mainly based on several publicly available packages: rasterio,



gdal[6], pyshp[7], geopandas[8], fiona[9], and xarray[10]. Complement code for generating any watershed's dataset will be released soon.

## 11 Conclusion

The dataset proposed in this paper provides a novel dataset for hydrological research in contiguous China. In the study area, there is no catchment attributes dataset has been proposed before, either a catchment-scale time series meteorological dataset. All catchments delaminated from the DEM are studied, covering contiguous China. The dataset includes daily meteorological forcing time-series data including precipitation, temperature, potential evapotranspiration, wind, ground surface temperature, pressure, humidity, sunshine duration and derived potential evapotranspiration of 4875 catchments. The proposed time series dataset is derived based on the quality-controlled site observation dataset, SURF_CLI_CHN_MUL_DAY. We will also release the complement code for generating any shapefile's meteorological time series within contiguous China based on the SURF_CLI_CHN_MUL_DAY dataset (freely available for Chinese researchers). The dataset has longer time series (from 1990 to 2018) and more meteorological variables than the previously proposed datasets. The dataset also includes 120+ catchment attributes, including soil, land cover, geology, climate indices and topography for each catchment. We produced a series of maps depicting the catchment attributes distributions in contiguous China. These maps present regional changes of various features; we also describe the relationships between them. The integration of multiple data sources into one dataset at a catchment-scale dramatically simplifies the data compilation process in research. Based on the dataset, we can test hypotheses and formulate valid conclusions under various conditions, not just limited to a few specific locations. Together with the Normal-Camels-YR dataset, the proposed dataset can help explore how different basin characteristics influence hydrological behaviours, learn the migration of hydrological behaviours between different basins, and to develop general frameworks for large-scale model evaluation and benchmarking in China.

## Appendix A: Modified Penman's equation

Penman's equation (Subramanya 2013), incorporating some modifications to the original formula, is:

$$PET = \frac{AH_n + E_a\gamma}{A + \gamma}$$

---

[6] https://github.com/OSGeo/gdal

[7] https://github.com/GeospatialPython/pyshp

[8] https://github.com/geopandas/geopandas

[9] https://github.com/Toblerity/Fiona

[10] https://github.com/pydata/xarray



where $PET$ is the daily potential evapotranspiration in mm per day; $A$ is the slope of the saturation vapour pressure ($ew$) vs temperature ($t$) curve at the mean air temperature, in mm of mercury per Celsius; $Hn$ is the net radiation in mm of evaporable water per day; $Ea$ is a parameter including wind speed and saturation deficit; $\gamma$ is the psychrometric constant = 0.49 mm of mercury per Celsius.

The relationship between $ew$ and $t$ is defined as:

$$e_w = 4.584 \exp\left(\frac{17.27t}{237.3 + t}\right)$$

The following equation estimates the net radiation:

$$H_n = H_a(1 - r)\left(a + b\frac{n}{N}\right) - \sigma T_a^4\left(0.56 - 0.092\sqrt{e_a}\right)\left(0.10 + 0.90\frac{n}{N}\right)$$

where $Ha$ is the incident solar radiation outside the atmosphere on a horizontal surface, expressed in mm of evaporable water per day (a function of the latitude and period of the year as indicated in Table A1); $a$ is a constant depending upon the latitude $\phi$ and is given by $a = 0.29\cos\phi$; $b$ is a constant = 0.52; $n$ is the sunshine duration in hours; $N$ is the maximum possible hours of bright sunshine (a function of latitude, see Table A2); $r$ is the reflection coefficient; $\sigma$ is the Stefan-Boltzman constant = $2.01 \times 10^{-9}$ mm/day; $Ta$ is the mean air temperature in degrees kelvin; $ea$ is the actual mean vapour pressure in the air in 470 mm of mercury.

**Table A1. Mean Monthly Solar Radiation, $Ha$ in mm of Evaporable Water/Day**

| North latitude | Jan | Feb | Mar | Apr | May | Jun | Jul | Aug | Sep | Oct | Nov | Dec |
|---|---|---|---|---|---|---|---|---|---|---|---|---|
| 0° | 14.5 | 15.0 | 15.2 | 14.7 | 13.9 | 13.4 | 13.5 | 14.2 | 14.9 | 15.0 | 14.6 | 14.3 |
| 10° | 12.8 | 13.9 | 14.8 | 15.2 | 15.0 | 14.8 | 14.8 | 15.0 | 14.9 | 14.1 | 13.1 | 12.4 |
| 20° | 10.8 | 12.3 | 13.9 | 15.2 | 15.7 | 15.8 | 15.7 | 15.3 | 14.4 | 12.9 | 11.2 | 10.3 |
| 30° | 8.5 | 10.5 | 12.7 | 14.8 | 16.0 | 16.5 | 16.2 | 15.3 | 13.5 | 11.3 | 9.1 | 7.9 |
| 40° | 6.0 | 8.3 | 11.0 | 13.9 | 15.9 | 16.7 | 16.3 | 14.8 | 12.2 | 9.3 | 6.7 | 5.4 |
| 50° | 3.6 | 5.9 | 9.1 | 12.7 | 15.4 | 16.7 | 16.1 | 13.9 | 10.5 | 7.1 | 4.3 | 3.0 |

The parameter $Ea$ is estimated as:

$$E_a = 0.35\left(1 + \frac{u_2}{160}\right)(e_w - e_a)$$

where $u2$ is the wind speed at $2m$ above ground in km/day; $ew$ is the saturation vapour pressure at mean air temperature in mm of mercury; $ea$ is the actual vapour pressure.

**Table A2. Mean Monthly Values of Possible Sunshine Hours, $N$**

| North latitude | Jan | Feb | Mar | Apr | May | Jun | Jul | Aug | Sep | Oct | Nov | Dec |
|---|---|---|---|---|---|---|---|---|---|---|---|---|
| 0° | 12.1 | 12.1 | 12.1 | 12.1 | 12.1 | 12.1 | 12.1 | 12.1 | 12.1 | 12.1 | 12.1 | 12.1 |





| 10° | 11.6 | 11.8 | 12.1 | 12.4 | 12.6 | 12.7 | 12.6 | 12.4 | 12.9 | 11.9 | 11.7 | 11.5 |
| 20° | 11.1 | 11.5 | 12.0 | 12.6 | 13.1 | 13.3 | 13.2 | 12.8 | 12.3 | 11.7 | 11.2 | 10.9 |
| 30° | 10.4 | 11.1 | 12.0 | 12.9 | 13.7 | 14.1 | 13.9 | 13.2 | 12.4 | 11.5 | 10.6 | 10.2 |
| 40° | 9.6 | 10.7 | 11.9 | 13.2 | 14.4 | 15.0 | 14.7 | 13.8 | 12.5 | 11.2 | 10.0 | 9.4 |
| 50° | 8.6 | 10.1 | 11.8 | 13.8 | 15.4 | 16.4 | 16.0 | 14.5 | 12.7 | 10.8 | 9.1 | 8.1 |

**Appendix B: Correlation analysis of catchment attributes**

To explore the potential connections between various types of watershed attributes, we did correlation analysis using the
480 Pearson correlation coefficient; the results can be found in Table B1, which shows the top five most relevant attributes for
each attribute, and the Fig. S1, the correlation matrix. The analysis result shows that the correlations between variables are
consistent with general understanding, justifying the rationality of the dataset:

(1) Subsurface permeability and porosity are highly correlated with geological attributes.

(2) LAI and NDVI have a high positive correlation (0.866).

(3) Root depth is most correlated with land cover types.

(4) In China, the savanna is mainly distributed in the southern coastal areas, resulting in that it is positively correlated with
average rainfall (0.604).

(5) Sand is positively correlated with saturated hydraulic conductivity (0.86) while the clay is negatively correlated (-0.763),
and catchments with a lot of rainfall are less likely to have soil with high hydraulic conductivity (-0.647).

(6) High altitude catchments tend to have lower saturated water content (-0.705).

**Table B1. The top five most relevant characteristics for each attribute (different soil layers for the same attribute are excluded, e.g. phihox_sl2 is not included in the top five most relevant attributes of phihox_sl1 though they are highly correlated)**

| Attribute | 1st | 2nd | 3rd | 4th | 5th |
|---|---|---|---|---|---|
| high_prec_freq | low_prec_dur(-0.58) | root_depth_50(-0.438) | root_depth_99(-0.436) | barren(-0.39) | pet_mean(-0.261) |
| high_prec_dur | elev(0.544) | theta_s_l6(-0.503) | prs_mean (-0.49) | theta_s_l5(-0.458) | rhu_mean(-0.431) |
| low_prec_freq | pre_mean(-0.881) | ssd_mean(0.841) | phihox_sl7(0.825) | phihox_sl6(0.818) | phihox_sl5(0.814) |
| low_prec_dur | barren(0.728) | rhu_mean(-0.723) | evp_mean(0.721) | ndvi_mean(-0.684) | root_depth_99(0.66) |
| frac_snow_daily | tem_mean(-0.951) | gst_mean(-0.949) | ssd_mean(0.777) | pre_mean(-0.762) | n_min(0.703) |
| p_seasonality | pre_mean(0.901) | rhu_mean(0.765) | ssd_mean(-0.764) | low_prec_freq(-0.712) | frac_snow_daily(-0.683) |
| pet_mean | cecsol_sl2(-0.66) | cecsol_sl1(-0.634) | cecsol_sl3(-0.628) | gst_mean(0.622) | bldfie_sl1(0.608) |
| pre_mean | p_seasonality(0.901) | low_prec_freq(-0.881) | ssd_mean(-0.858) | rhu_mean(0.832) | phihox_sl7(-0.819) |
| tem_mean | gst_mean(0.992) | frac_snow_daily(-0.951) | pre_mean(0.747) | ssd_mean(-0.709) | p_seasonality(0.681) |
| prs_mean | elev(-0.889) | e_max(0.707) | lon(0.707) | e_min(0.707) | rhu_mean(0.603) |



| | | | | |
|---|---|---|---|---|
| rhu_mean | ssd_mean(-0.887) | pre_mean(0.832) | evp_mean(-0.823) | ndvi_mean(0.813) | low_prec_freq(-0.803) |
| evp_mean | ndvi_mean(-0.845) | rhu_mean(-0.823) | ssd_mean(0.756) | e_min(-0.731) | lon(-0.73) |
| win_mean | ssd_mean(0.581) | frac_snow_daily(0.571) | tem_mean(-0.52) | gst_mean(-0.507) | low_prec_freq(0.477) |
| ssd_mean | rhu_mean(-0.887) | pre_mean(-0.858) | low_prec_freq(0.841) | frac_snow_daily(0.777) | p_seasonality(-0.764) |
| gst_mean | tem_mean(0.992) | frac_snow_daily(-0.949) | pre_mean(0.743) | n_min(-0.693) | lat(-0.693) |
| geol_permeability | ss(-0.408) | sm(-0.403) | su(0.399) | sc(0.323) | bdticm(0.24) |
| geol_porosity | su(0.627) | pa(-0.575) | phihox_sl1(0.46) | phihox_sl3(0.454) | phihox_sl4(0.453) |
| ig | snow and ice(0.471) | tksatu_l5(0.324) | tksatu_l3(0.318) | tksatu_l4(0.306) | tksatu_l2(0.275) |
| pa | geol_porosity(-0.575) | phihox_sl1(-0.314) | phihox_sl3(-0.302) | phihox_sl2(-0.301) | phihox_sl4(-0.297) |
| sc | geol_porosity(-0.362) | geol_permeability(0.323) | n_max(-0.317) | lat(-0.317) | n_min(-0.316) |
| su | geol_porosity(0.627) | bdticm(0.599) | cropland(0.468) | phihox_sl1(0.44) | phihox_sl4(0.439) |
| sm | geol_permeability(-0.403) | su(-0.385) | cropland(-0.268) | bdticm(-0.233) | e_max(-0.228) |
| vi | deciduous broadleaf tree(0.214) | geol_porosity(-0.18) | lai_max(0.165) | lai_dif(0.159) | e_max(0.157) |
| mt | geol_porosity(-0.412) | evergreen needleleaf tree(0.327) | orcdrc_sl3(0.265) | orcdrc_sl4(0.258) | bldfie_sl5(-0.254) |
| ss | geol_permeability(-0.408) | su(-0.287) | sm(-0.206) | geol_porosity(0.2) | tksatu_l6(-0.156) |
| pi | deciduous broadleaf tree(0.299) | geol_porosity(-0.208) | e_max(0.161) | lon(0.161) | e_min(0.16) |
| va | geol_porosity(-0.218) | high_prec_dur(0.191) | tem_mean(-0.167) | gst_mean(-0.16) | su(-0.16) |
| wb | water bodies(0.674) | permanent wetland(0.379) | root_depth_50(-0.164) | theta_s_l3(0.148) | theta_s_l4(0.147) |
| pb | theta_s_l6(-0.137) | theta_s_l5(-0.133) | elev(m)(0.124) | theta_s_l4(-0.114) | prs_mean(-0.102) |
| vb | cecsol_sl2(0.222) | cecsol_sl3(0.213) | cecsol_sl1(0.212) | cecsol_sl4(0.211) | cecsol_sl5(0.208) |
| nd | snow and ice(0.206) | theta_s_l2(-0.154) | theta_s_l3(-0.151) | theta_s_l1(-0.144) | tksatu_l4(0.136) |
| py | phihox_sl1(-0.214) | phihox_sl2(-0.207) | phihox_sl3(-0.207) | phihox_sl4(-0.205) | phihox_sl5(-0.202) |
| ev | tksatu_l3(0.07) | tksatu_l4(0.066) | barren(0.064) | tksatu_l2(0.061) | tksatu_l1(0.061) |
| lai_dif | ndvi_mean(0.866) | phihox_sl4(-0.809) | phihox_sl2(-0.807) | phihox_sl5(-0.807) | phihox_sl6(-0.807) |
| lai_max | ndvi_mean(0.856) | phihox_sl4(-0.815) | phihox_sl5(-0.814) | phihox_sl6(-0.814) | phihox_sl2(-0.813) |
| ndvi_mean | lai_dif(0.866) | lai_max(0.856) | evp_mean(-0.845) | rhu_mean(0.813) | barren(-0.772) |
| root_depth_50 | barren(0.856) | low_prec_dur(0.626) | grassland(-0.537) | ndvi_mean(-0.513) | evp_mean(0.497) |



| | | | | |
|---|---|---|---|---|
| root_depth_99 | barren(0.897) | low_prec_dur(0.66) | ndvi_mean(-0.628) | evp_mean(0.604) | rhu_mean(-0.486) |
| evergreen needleleaf tree | slope(0.398) | bldfie_sl4(-0.391) | bldfie_sl5(-0.384) | bldfie_sl3(-0.372) | bldfie_sl7(-0.366) |
| evergreen broadleaf tree | pre_mean(0.504) | lai_max(0.483) | phihox_sl7(-0.477) | lai_dif(0.471) | phihox_sl6(-0.47) |
| deciduous needleleaf tree | woody savanna(0.241) | cecsol_sl2(0.231) | orcdrc_sl2(0.226) | pet_mean(-0.215) | bldfie_sl1(-0.214) |
| deciduous broadleaf tree | lai_max(0.459) | lai_dif(0.452) | cecsol_sl1(0.433) | bldfie_sl1(-0.413) | e_max(0.361) |
| mixed forest | orcdrc_sl1(0.501) | lai_max(0.471) | lai_dif(0.466) | phihox_sl6(-0.462) | phihox_sl7(-0.461) |
| closed shrubland | theta_s_l1(-0.084) | grav(0.079) | sc(0.075) | theta_s_l2(-0.072) | urban and built-up land(0.064) |
| open shrubland | high_prec_dur(0.155) | theta_s_l6(-0.151) | rhu_mean(-0.149) | prs_mean(-0.147) | evp_mean(0.139) |
| woody savanna | lai_max(0.633) | lai_dif(0.631) | phihox_sl7(-0.592) | phihox_sl6(-0.59) | phihox_sl5(-0.585) |
| savanna | pre_mean(0.604) | phihox_sl7(-0.55) | clay(0.547) | phihox_sl6(-0.543) | phihox_sl5(-0.537) |
| grassland | root_depth_50(-0.537) | tem_mean(-0.496) | gst_mean(-0.491) | frac_snow_daily(0.469) | phihox_sl6(0.438) |
| permanent wetland | wb(0.379) | water bodies(0.349) | p_seasonality(0.3) | pre_mean(0.248) | pop_dnsty(0.23) |
| cropland | su(0.468) | lon(0.412) | e_min(0.412) | e_max(0.412) | elev(-0.388) |
| urban and built-up land | pop_dnsty(0.811) | pop(0.399) | p_seasonality(0.286) | tem_mean(0.261) | elev(-0.244) |
| cropland/natural vegetaion | ssd_mean(-0.458) | savanna(0.381) | rhu_mean(0.371) | frac_snow_daily(-0.367) | tem_mean(0.364) |
| snow and ice | tksatu_l5(0.568) | tksatu_l3(0.561) | tksatu_l4(0.533) | tksatu_l2(0.506) | tksatu_l1(0.503) |
| barren | root_depth_99(0.897) | root_depth_50(0.856) | ndvi_mean(-0.772) | low_prec_dur(0.728) | evp_mean(0.698) |
| water bodies | wb(0.674) | permanent wetland(0.349) | root_depth_50(-0.192) | root_depth_99(-0.154) | theta_s_l3(0.153) |
| length | area(0.849) | circulatory_ratio(-0.491) | elongation_ratio(-0.451) | form_factor(-0.436) | compactness_coefficient(0.292) |
| area | length(0.849) | pop(0.418) | circulatory_ratio(-0.255) | cecsol_sl1(0.142) | bldfie_sl2(-0.138) |
| form_factor | elongation_ratio(0.992) | circulatory_ratio(0.647) | shape_factor(-0.506) | length(-0.436) | compactness_coefficient(-0.383) |
| shape_factor | compactness_coefficient(0.786) | elongation_ratio(-0.566) | form_factor(-0.506) | circulatory_ratio(-0.372) | length(0.266) |
| compactness_coefficient | shape_factor(0.786) | circulatory_ratio(-0.594) | elongation_ratio(-0.421) | form_factor(-0.383) | length(0.292) |





| circulatory_ratio | elongation_ratio(0.651) | form_factor(0.647) | compactness_coefficient(-0.594) | length(-0.491) | shape_factor(-0.372) |
|---|---|---|---|---|---|
| elongation_ratio | form_factor(0.992) | circulatory_ratio(0.651) | shape_factor(-0.566) | length(-0.451) | compactness_coefficient(-0.421) |
| elev(m) | prs_mean(-0.889) | e_min(-0.753) | lon(-0.752) | e_max(-0.752) | theta_s_l4(-0.7) |
| slope(m/km) | n_min(-0.552) | lat(-0.551) | n_max(-0.55) | phihox_sl7(-0.491) | orcdrc_sl1(0.49) |
| n_min | lat(1.0) | frac_snow_daily(0.703) | gst_mean(-0.693) | pre_mean(-0.651) | tem_mean(-0.648) |
| n_max | lat(1.0) | frac_snow_daily(0.701) | gst_mean(-0.692) | pre_mean(-0.65) | tem_mean(-0.647) |
| e_min | lon(1.0) | elev(-0.753) | evp_mean(-0.731) | prs_mean(0.707) | ndvi_mean(0.691) |
| e_max | lon(1.0) | elev(-0.752) | evp_mean(-0.729) | prs_mean(0.707) | ndvi_mean(0.69) |
| pop(people) | area(0.418) | urban and built-up land(0.399) | tem_mean(0.318) | p_seasonality(0.317) | frac_snow_daily(-0.304) |
| pop_dnsty(people/km$^2$) | urban and built-up land(0.811) | p_seasonality(0.426) | tem_mean(0.412) | gst_mean(0.395) | frac_snow_daily(-0.39) |
| lon | e_max(1.0) | e_min(1.0) | elev(-0.752) | evp_mean(-0.73) | prs_mean(0.707) |
| lat | n_min(1.0) | n_max(1.0) | frac_snow_daily(0.702) | gst_mean(-0.693) | pre_mean(-0.651) |
| tksatu_l1 | snow and ice(0.503) | silt(-0.465) | som(-0.366) | sand(0.362) | log_k_s_l5(0.327) |
| tksatu_l2 | snow and ice(0.506) | silt(-0.49) | sand(0.406) | som(-0.365) | log_k_s_l5(0.364) |
| tksatu_l3 | snow and ice(0.561) | silt(-0.489) | sand(0.409) | ndvi_mean(-0.368) | clay(-0.334) |
| tksatu_l4 | snow and ice(0.533) | silt(-0.49) | sand(0.465) | ndvi_mean(-0.455) | log_k_s_l5(0.414) |
| tksatu_l5 | snow and ice(0.568) | silt(-0.402) | ndvi_mean(-0.375) | sand(0.348) | lai_dif(-0.326) |
| tksatu_l6 | snow and ice(0.449) | bdticm(0.403) | log_k_s_l6(0.384) | su(0.38) | low_prec_freq(0.36) |
| log_k_s_l1 | sand(0.858) | clay(-0.733) | pre_mean(-0.553) | phihox_sl7(0.551) | phihox_sl6(0.546) |
| log_k_s_l2 | sand(0.86) | clay(-0.729) | phihox_sl7(0.575) | phihox_sl6(0.569) | pre_mean(-0.568) |
| log_k_s_l3 | sand(0.859) | clay(-0.728) | pre_mean(-0.571) | phihox_sl7(0.571) | phihox_sl6(0.565) |
| log_k_s_l4 | sand(0.82) | clay(-0.752) | pre_mean(-0.647) | phihox_sl7(0.636) | phihox_sl6(0.63) |
| log_k_s_l5 | sand(0.773) | clay(-0.714) | phihox_sl7(0.654) | phihox_sl6(0.649) | phihox_sl5(0.646) |
| log_k_s_l6 | sand(0.688) | clay(-0.687) | phihox_sl7(0.665) | phihox_sl6(0.662) | pre_mean(-0.662) |
| theta_s_l1 | grav(-0.705) | elev(-0.422) | rhu_mean(0.407) | clay(0.401) | pdep(0.4) |
| theta_s_l2 | grav(-0.713) | elev(-0.505) | pdep(0.475) | e_min(0.442) | lon(0.441) |
| theta_s_l3 | grav(-0.662) | elev(-0.638) | prs_mean(0.554) | pdep(0.52) | e_min(0.516) |
| theta_s_l4 | elev(-0.7) | grav(-0.663) | prs_mean(0.594) | pdep(0.571) | e_min(0.51) |
| theta_s_l5 | elev(-0.656) | grav(-0.584) | prs_mean(0.536) | pdep(0.501) | rhu_mean(0.467) |
| theta_s_l6 | elev(-0.637) | prs_mean(0.525) | grav(-0.513) | high_prec_dur(-0.503) | rhu_mean(0.475) |
| orcdrc_sl7 | cecsol_sl2(0.758) | bldfie_sl2(-0.745) | bldfie_sl4(-0.744) | bldfie_sl1(-0.737) | cecsol_sl3(0.735) |
| orcdrc_sl3 | bldfie_sl2(-0.876) | bldfie_sl4(-0.875) | bldfie_sl3(-0.874) | bldfie_sl5(-0.849) | bldfie_sl1(-0.848) |
| orcdrc_sl4 | bldfie_sl4(-0.823) | bldfie_sl2(-0.809) | bldfie_sl3(-0.803) | bldfie_sl5(-0.803) | bldfie_sl1(-0.787) |





| orcdrc_sl5 | bldfie_sl4(-0.759) | bldfie_sl2(-0.754) | bldfie_sl5(-0.745) | bldfie_sl1(-0.745) | bldfie_sl3(-0.731) |
|---|---|---|---|---|---|
| orcdrc_sl6 | cecsol_sl2(0.733) | bldfie_sl4(-0.733) | bldfie_sl2(-0.728) | bldfie_sl1(-0.725) | bldfie_sl5(-0.721) |
| orcdrc_sl2 | bldfie_sl2(-0.917) | bldfie_sl1(-0.908) | bldfie_sl3(-0.861) | cecsol_sl1(0.854) | bldfie_sl4(-0.854) |
| orcdrc_sl1 | phihox_sl2(-0.826) | phihox_sl1(-0.824) | phihox_sl3(-0.822) | phihox_sl4(-0.819) | phihox_sl5(-0.813) |
| phihox_sl7 | low_prec_freq(0.825) | pre_mean(-0.819) | lai_max(-0.806) | orcdrc_sl1(-0.804) | lai_dif(-0.799) |
| phihox_sl6 | low_prec_freq(0.818) | lai_max(-0.814) | pre_mean(-0.81) | orcdrc_sl1(-0.807) | lai_dif(-0.807) |
| phihox_sl5 | lai_max(-0.814) | low_prec_freq(0.814) | orcdrc_sl1(-0.813) | lai_dif(-0.807) | pre_mean(-0.801) |
| phihox_sl4 | orcdrc_sl1(-0.819) | lai_max(-0.815) | lai_dif(-0.809) | low_prec_freq(0.804) | pre_mean(-0.781) |
| phihox_sl3 | orcdrc_sl1(-0.822) | lai_max(-0.813) | lai_dif(-0.806) | low_prec_freq(0.799) | pre_mean(-0.772) |
| phihox_sl2 | orcdrc_sl1(-0.826) | lai_max(-0.813) | lai_dif(-0.807) | low_prec_freq(0.798) | pre_mean(-0.767) |
| phihox_sl1 | orcdrc_sl1(-0.824) | lai_max(-0.804) | lai_dif(-0.798) | low_prec_freq(0.78) | pre_mean(-0.741) |
| bldfie_sl7 | orcdrc_sl3(-0.775) | orcdrc_sl4(-0.747) | orcdrc_sl5(-0.698) | orcdrc_sl2(-0.698) | orcdrc_sl6(-0.671) |
| bldfie_sl6 | orcdrc_sl3(-0.776) | orcdrc_sl4(-0.748) | orcdrc_sl5(-0.701) | orcdrc_sl2(-0.694) | orcdrc_sl6(-0.677) |
| bldfie_sl5 | orcdrc_sl3(-0.849) | orcdrc_sl2(-0.81) | orcdrc_sl4(-0.803) | orcdrc_sl5(-0.745) | orcdrc_sl7(-0.728) |
| bldfie_sl4 | orcdrc_sl3(-0.875) | orcdrc_sl2(-0.854) | orcdrc_sl4(-0.823) | cecsol_sl1(-0.763) | orcdrc_sl5(-0.759) |
| bldfie_sl1 | orcdrc_sl2(-0.908) | cecsol_sl1(-0.891) | orcdrc_sl3(-0.848) | cecsol_sl2(-0.828) | orcdrc_sl4(-0.787) |
| bldfie_sl3 | orcdrc_sl3(-0.874) | orcdrc_sl2(-0.861) | orcdrc_sl4(-0.803) | cecsol_sl1(-0.795) | som(-0.787) |
| bldfie_sl2 | orcdrc_sl2(-0.917) | orcdrc_sl3(-0.876) | cecsol_sl1(-0.87) | orcdrc_sl4(-0.809) | som(-0.808) |
| cecsol_sl1 | bldfie_sl1(-0.891) | bldfie_sl2(-0.87) | orcdrc_sl2(0.854) | bldfie_sl3(-0.795) | orcdrc_sl3(0.781) |
| cecsol_sl2 | bldfie_sl1(-0.828) | orcdrc_sl2(0.822) | bldfie_sl2(-0.798) | orcdrc_sl7(0.758) | orcdrc_sl3(0.746) |
| cecsol_sl5 | bldfie_sl1(-0.681) | orcdrc_sl2(0.664) | orcdrc_sl7(0.649) | bldfie_sl2(-0.645) | orcdrc_sl6(0.636) |
| cecsol_sl4 | bldfie_sl1(-0.72) | orcdrc_sl2(0.717) | orcdrc_sl7(0.693) | bldfie_sl2(-0.692) | orcdrc_sl6(0.679) |
| cecsol_sl3 | bldfie_sl1(-0.784) | orcdrc_sl2(0.776) | bldfie_sl2(-0.76) | orcdrc_sl7(0.735) | orcdrc_sl3(0.733) |
| cecsol_sl7 | bldfie_sl1(-0.661) | orcdrc_sl7(0.654) | orcdrc_sl2(0.642) | orcdrc_sl6(0.64) | orcdrc_sl5(0.619) |
| cecsol_sl6 | bldfie_sl1(-0.648) | orcdrc_sl2(0.637) | orcdrc_sl7(0.632) | orcdrc_sl6(0.62) | bldfie_sl2(-0.61) |
| bdticm | su(0.599) | low_prec_freq(0.463) | log_k_s_l6(0.439) | phihox_sl2(0.437) | phihox_sl7(0.436) |
| pdep | elev(-0.662) | theta_s_l4(0.571) | e_min(0.566) | lon(0.565) | e_max(0.564) |
| por | silt(0.573) | clay(0.366) | tksatu_l2(-0.317) | som(0.314) | tksatu_l1(-0.309) |
| clay | pre_mean(0.763) | log_k_s_l4(-0.752) | log_k_s_l1(-0.733) | log_k_s_l2(-0.729) | log_k_s_l3(-0.728) |



| sand | log_k_s_l2(0.86) | log_k_s_l3(0.859) | log_k_s_l1(0.858) | log_k_s_l4(0.82) | log_k_s_l5(0.773) |
|------|------------------|-------------------|-------------------|------------------|-------------------|
| silt | por(0.573) | sand(-0.558) | log_k_s_l3(-0.557) | log_k_s_l2(-0.547) | log_k_s_l1(-0.545) |
| grav | theta_s_l2(-0.713) | theta_s_l1(-0.705) | theta_s_l4(-0.663) | theta_s_l3(-0.662) | theta_s_l5(-0.584) |
| som | bldfie_sl2(-0.808) | bldfie_sl3(-0.787) | bldfie_sl1(-0.759) | bldfie_sl4(-0.747) | orcdrc_sl2(0.74) |

**Financial support**

This research has been supported by the National Key Research and Development Program (2018YFC0407901,
2018YFC0407905), the National Natural Science Fund of China (51779100), and the Central Public-interest Scientific
Institution Basal Research Fund (HKY-JBYW-2020-21, HKY-JBYW-2020-07).

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
