# Peer review of "CCAM: China Catchment Attributes and Meteorology dataset"

_Earth System Science Data, 2021_

## Author Comment (AC5)

Date: 2021-08-10
Manuscript Number: ESSD-2021-71
Title of Article: Catchment attributes and meteorology for large sample study in contiguous China
Name of the Author: Zhen Hao
Email Address of the Author: zhen.hao18@alumni.imperial.ac.uk

Dear Editor,
Dear Referees,

Both reviewers gave very pertinent suggestions, and we thank them for their efforts to review this article. After being reviewed, we re-reviewed our work and re-examined its contribution to the entire hydrological community. We are writing to describe how we have revised the paper and the parts we have emphasized and clarified in the article.

We want to restate the importance of this work: though streamflow data is critical for large-sample hydrology research, large-scale catchment characteristics data are also vital. As stated in the initial CAMELS paper (which calculated the attributes, but the streamflow data was released in previous work):

"Although catchment attributes are routinely used when working with a handful of catchments, there is a growing recognition that a large sample of catchments can provide insights that cannot be gained from a small sample. Large-sample data sets enable us to concentrate on catchment similarities and on the formulation of conclusions that are valid for a large number of (gauged and ungauged) catchments. Individual catchments can then be considered to be part of a continuum of catchment attributes, which vary in space along several gradients (such as aridity or soil depth). Working with a large number of catchments enables us to study changes along different gradients and to better disentangle the effects of catchment attributes on catchment behavior. This is particularly useful for comparative hydrology, i.e., to identify how similarities and differences between locations influence ecohydrological processes. Further, large-sample hydrology opens new opportunities for data analysis and, for instance, makes it possible to explore interrelationships between catchment attributes on the basis of their spatial patterns, as exemplified later in this study using map comparisons."

There will be no large-scale streamflow data sharing within China in the foreseeable future, but Chinese researchers do have large-scale streamflow data. What hinders large-sample hydrology research is well-organized basin attributes data. The organization of the data into a catchment scale is the first of this kind in China. This work will also help researchers in other regions calculate watershed attributes by referencing the public code, especially attributes that are not trivial to calculate (e.g., p_seasonality). **As far as we know, there are already researches based on our data**.

To further improve the usability and influence of the code, we have reformulated the code such that the user can generate a basin's characteristics with just a "one-click" when the required source data

are prepared, which will significantly improve the accessibility of the catchment attributes.

Due to the strict redistribution policy of streamflow data, we are afraid not to be able to release the original streamflow data and the mean streamflow. We must ensure that the source data is not released, but we want the released data to be useful, so we present the current solution. The current data can be used in such a situation: when it is desirable to verify the generalization ability of a machine learning model on a global scale, HydroMLYR (new name) can support the verification of the performance in the Yellow River Basin.

There are also some other shortcomings that can be fixed. Next, we respond to these questions one by one:

(1) **RC**: Wrong citation style when the citation appears at the beginning of a sentence. **AC**: "(Kratzert, Klotz et al. 2019) shows" should be "Kratzert, Klotz et al. (2019) shows" and similar for others. **Changes**: L55, L60, L71, L72, L74, L193, L196, L198

(2) **RC**: Texts in Figures 3,4,5 are hardly readable. **AC**: We have fixed it by redrawing these figures. **Changes**: Figures 3,4,5

(3) **RC**: The irrationality of using the Pearsons Correlation Coefficient to access the correlation between catchment attributes. **AC**: Although the Pearson's correlation can only provide a complete description of the association when both the two variables are standard, we think the most doubtful part of using it is that it assumes a linear relationship (a change in one variable will cause a proportional change to the other). Because there are too many variables, we can't plot the scatterplot one by one to check whether the relationships are linear. We think Spearman or Kendall's Tau may be more suitable in this case due to its wider application range. Even if the relationship between the two variables is linear, Spearman or Kendall can also return a very close result to Pearson. However, if the relationship between two variables is only monotonic, Pearson will have information loss. **Changes**: L595-L615 (Appendix B)

(4) **RC**: The suggestion of not using the name "CAMELS". **AC**: The paper title sounds like the title of a CAMELS dataset, and the use of "Normal-Camels-YR" might be misleading. We suggest using CCAM to stand for "China Catchment Attributes and Meteorology dataset" and HydroMLYR to stand for "Hydrology dataset for Machine Learning of the Yellow River Basin." The new names may avoid readers' wrong expectation of the data set and more clearly indicates the purpose. **Changes**: L1

(5) **RC**: SURF_CLI_CHN_MUL_DAY is only freely available for Chinese researchers (L444). This is a non-negligible constraint. Furthermore, I don't see a paper documenting the SURF_CLI_CHN_MUL_DAY dataset and the link provided (http://data.cma.cn/data/cdcdetail/dataCode/SURF_CLI_CHN_MUL_DAY.html) leads to a page in Chinese. **AC**: We are sorry that we made a false statement about the facts at the beginning. We found on the registration page that foreign researchers can also register, but the interface is still in Chinese, which is out of our control. The SURF_CLI_CHN_MUL_DAY data was issued by the National

Meteorological Information Center of the China Meteorological Administration (NMIC/CMA). The data is quality controlled, and it is widely used in research in China. However, it does not have a related paper. **Changes**: L435

(6) **RC**: Some decisions made by the authors are puzzling. **AC**: In fact, the location of hydrological observation stations can be observed through remote sensing satellite images and then combined with HydroSHEDS's River network to determine their location. Then the boundaries of the basin can be determined based on the publicly available DEM, but we cannot release the names of these hydrological observation stations; these are sensitive information. **Changes**: We do not think it is necessary to explain this matter in the article.

In the past few days, we have made extensive efforts to reorganize our code. Combined with the data set that has been released, we are aiming to achieve two goals:
(1) Researchers can quickly obtain catchment attributes and meteorological time series of the local catchment from our data set.
(2) If the local catchment has a custom boundary, using our code can calculate the catchment attributes and meteorological time series quickly based on the given boundary. Our code currently supports one-click generation of all static attributes, as long as the required source data has been prepared according to the instruction, and the generation of the meteorological time series can also be quite effortless.

https://github.com/haozhen315/CCAM-China-Catchment-Attributes-and-Meteorology-dataset

Best regards,
Zhen Hao

---

## Author Response (AR2)

<h1 style="text-align:center">Author's General Response</h1>

**Date: 2021-08-16**

**Manuscript Number: ESSD-2021-71**

**Title of Article: CCAM: China Catchment Attributes and Meteorology dataset**

**Name of the Author: Zhen Hao**

**Email Address of the Author: zhen.hao18@alumni.imperial.ac.uk**

Dear Editor,

Dear Referees,

Both reviewers gave very pertinent suggestions, and we thank them for their efforts to review this article. After being reviewed, we re-reviewed our work and re-examined its contribution to the entire hydrological community. We are writing to describe how we have revised the paper and the parts we have emphasized and clarified in the article.

We want to restate the importance of this work: though streamflow data is critical for large-sample hydrology research, large-scale catchment characteristics data are also vital. As stated in the initial CAMELS paper (which calculated the attributes, but the streamflow data was released in previous work):

"Although catchment attributes are routinely used when working with a handful of catchments, there is a growing recognition that a large sample of catchments can provide insights that cannot be gained from a small sample. Large-sample data sets enable us to concentrate on catchment similarities and on the formulation of conclusions that are valid for a large number of (gauged and ungauged) catchments. Individual catchments can then be considered to be part of a continuum of catchment attributes, which vary in space along several gradients (such as aridity or soil depth). Working with a large number of catchments enables us to study changes along different gradients and to better disentangle the effects of catchment attributes on catchment behavior. This is particularly useful for comparative hydrology, i.e., to identify how similarities and differences between locations influence ecohydrological processes. Further, large-sample hydrology opens new opportunities for data

analysis and, for instance, makes it possible to explore interrelationships between catchment attributes on the basis of their spatial patterns, as exemplified later in this study using map comparisons."

There will be no large-scale streamflow data sharing within China in the foreseeable future, but Chinese researchers do have large-scale streamflow data. What hinders large-sample hydrology research is well-organized basin attributes data. The organization of the data into a catchment scale is the first of this kind in China. This work will also help researchers in other regions calculate watershed attributes by referencing the public code, especially attributes that are not trivial to calculate (e.g., p_seasonality). As far as we know, there are already research based on our data.

To further improve the usability and influence of the code, we have reformulated the code such that the user can generate a basin's characteristics with just a "one-click" when the required source data are prepared, which will significantly improve the accessibility of the catchment attributes.

**Highlight changes in the revised manuscript:**

(1) Abstract has been rewritten to avoid readers' wrong expectations for the streamflow data.

(2) The paper's name has been changed to "CCAM: China Catchment Attributes and Meteorology dataset".

(3) In the Introduction section, some adjustments have been made to the content sequence. First, the large-scale hydrological data set and its progress are introduced. After the YRB data set is introduced, the data-driven method is briefly introduced.

(4) All figured are redrawn to improve readability and aesthetics.

(5) A more detailed introduction to the Yellow River data set, including a description of the average discharge record length, etc.

(6) Three Appendixes are added:

1. A description of the data source and data processing process.
2. A description of the basin boundary data set used.
3. The accompany guidelines for the code that can generate basin attributes for a given basin boundary.

Next, we respond to the referee questions one by one:

**Point-by-point reply to Referee 1**

**RC**: However, at this time, the source code is not yet available. If possible, however, the source code should already be published along with the data, which would add significant value to the paper.

**AC**: The code is now available at: https://github.com/haozhen315/CCAM-China-Catchment-Attributes-and-Meteorology-dataset

To further improve the usability and influence of the code, we have reformulated the code such that the user can generate a basin's characteristics with just a "one-click" when the required source data are prepared, which will significantly improve the accessibility of the catchment attributes.

In the past few days, we have made extensive efforts to reorganize our code. Combined with the data set that has been released, we are aiming to achieve two goals:

(1) Researchers can quickly obtain catchment attributes and meteorological time series of the local catchment from our data set.

(2) If the local catchment has a custom boundary, using our code can calculate the catchment attributes and meteorological time series quickly based on the given boundary. Our code currently supports one-click generation of all static attributes, as long as the required source data has been prepared according to the instruction, and the generation of the meteorological time series can also be quite effortless.

**RC**: Many of the normalized discharge time series seem to have gaps, which reduces their usability for further analysis. It would be useful for the reader if the mean length of complete discharge time series could be quantified, or the gaps could be addressed in general

**AC**: The original streamflow observations are not continuous. The average record length is 11.3 years. In the revised version of the dataset, we separately provide continuous streamflow observations with an average record length of 8.3 years.

**RC**: Also, the reason for the normalization should be addressed; as from the normalized time series, e.g. no mean annual discharge can be derived; and this parameter is also not given in the metadata of the catchments.

**AC**: Due to the strict redistribution policy of streamflow data, we are afraid not to be able to release the original streamflow data and the mean discharge. We must ensure that the source data is not released, but we want the released data to be useful, so we present the current solution. The current data can be used in such a situation: when it is desirable to verify the generalization ability of a machine learning model on a global scale, HydroMLYR (new name) can support the verification of the performance in the Yellow River Basin.

**RC**: line 28: Either name the main processes of the hydrological cycle, or focus on specific processes in terrestrial catchments. Careful with the right terms: rainfall instead of raindrop

**AC**: This sentence has been reformulated as "Rainfall, interception, evaporation and evapotranspiration, groundwater flow, subsurface flow and surface runoff are the main components of the terrestrial hydrological cycle."

**RC**: line 33: "[..] it is possible for the hydrological model to learn [..]": this is only true for a spacial type of models (machine-learning models)

**AC**: This sentence has been reformulated as "However, by examining a large sample of catchments, it is possible for a data-driven model to learn the similarities and differences of hydrological behaviours across catchments (Kratzert, Klotz et al. 2019)."

**RC**: line 37: citation style: instead of "(Kratzert, Klotz et al. 2019) shows" it should be "Kratzert, Klotz et al. (2019) shows". also applies for similar citations later in the text

**AC**: "(Kratzert, Klotz et al. 2019) shows" should be "Kratzert, Klotz et al. (2019) shows". Other similar issues have also been fixed.

**Changes**: L55, L60, L71, L72, L74, L193, L196, L198

**RC**: Figures 3,4,5: as it is already written in a community comment, the text in the figures is hardly readable.

**AC**: We have fixed it by redrawing these figures.

**Changes**: Figures 3,4,5

**RC**: Appendix B: Assumption for the use of Pearsons Correlation Coefficient is normal distribution; and linear relationship is assumed. is this really always the case here?

**AC**: Although the Pearson's correlation can only provide a complete description of the association when both the two variables are standard, we think the most doubtful part of using it is that it assumes a linear relationship (a change in one variable will cause a proportional change to the other). Because there are too many variables, we can't plot the scatterplot one by one to check whether the relationships are linear. We think Spearman or Kendall's Tau may be more suitable in this case due to its wider application range. Even if the relationship between the two variables is linear, Spearman or Kendall can also return a very close result to Pearson. However, if the relationship between two variables is only monotonic, Pearson will have information loss.

**Changes**: L595-L615 (Appendix B)

**Point-by-point reply to Referee 2**

**RC**: In the abstract, the authors state their "dataset provides numerous opportunities for comparative hydrological research, such as examining the difference in hydrological behaviours across different catchments and building general rainfall-runoff modelling frameworks for many catchments instead of limited to a few". My concern is that the scope of this dataset (in its current form) might be more limited, because of the following restrictions imposed to the streamflow data: […]

**AC**: Due to the strict redistribution policy of streamflow data, we are afraid not to be able to release the original streamflow data and the mean streamflow. We must ensure that the source data is not released, but we want the released data to be useful, so we present the current solution. The current data can be used in such a situation: when it is desirable to verify the generalization ability of a machine learning model on a global scale, HydroMLYR (new name) can support the verification of the performance in the Yellow River Basin. The abstract has also been rewritten to avoid those claims.

**RC**: I understand that releasing the true streamflow time series is challenging, but some decisions made by the authors are puzzling. For instance, "for confidentiality, the names of these basins have not been announced", but shapefiles are provided and give (presumably) the exact location of the catchments. Likewise, the mean streamflow can be inferred quite readily from catchment descriptors, which makes me feel that the normalisation of the timeseries is unnecessary.

**AC**: In fact, the location of hydrological observation stations can be observed through remote sensing satellite images and then combined with HydroSHEDS's River network to determine their location. Then the boundaries of the basin can be determined based on the publicly available DEM, but we cannot release the names of these hydrological observation stations; these are sensitive information. In addition, indicators that can be used to calculate the mean discharge have not been released.

**RC**: Hence, I recommend that the authors do not use the name CAMELS, as all the CAMELS datasets provide daily streamflow timeseries for their hundreds of catchments, which many in the community see as their most important characteristic. There are many alternative naming options, including exotic animal names (as illustrated by the recent LamaH dataset - https://essd.copernicus.org/preprints/essd-2021-72/).

**AC**: The paper title sounds like the title of a CAMELS dataset, and the use of "Normal-Camels-YR" might be misleading. We suggest using CCAM to stand for "China Catchment Attributes and Meteorology dataset" and HydroMLYR to stand for "Hydrology dataset for Machine Learning of the Yellow River Basin." The new names may avoid readers' wrong expectation of the data set and more clearly indicates the purpose.

**Changes**: L1

**RC**: Data availability/reproducibility: The abstract mentions that "complement code for generating the dataset will be open-sourced such that the user can generate meteorological series and catchment attributes for any watershed within contiguous China", yet the conclusion makes it clear that the forcing dataset SURF_CLI_CHN_MUL_DAY is only freely available for Chinese researchers (L444). This is a non-negligible constraint. Furthermore, I don't see a paper documenting the SURF_CLI_CHN_MUL_DAY dataset, and the link provided (http://data.cma.cn/data/cdcdetail/dataCode/SURF_CLI_CHN_MUL_DAY.html) leads to page in Chinese.

**AC**: We are sorry that we made a false statement about the facts at the beginning. We found on the registration page that foreign researchers can also register, but the interface is still in Chinese, which is out of our control. The SURF_CLI_CHN_MUL_DAY data was issued by the National Meteorological Information Center of the China Meteorological Administration (NMIC/CMA). The data is quality controlled, and it is widely used in research in China. However, it does not have a related paper.

**Changes**: L435

Best,

Zhen Hao

---

## Author Response (AR3)

**Date:** 22nd of Oct 2021
**Manuscript Number:** ESSD-2021-71
**Title of Article:** CCAM: China Catchment Attributes and Meteorology dataset
**Name of the Corresponding Author:** Zhen Hao
**Email Address of the Corresponding Author:** zhen.hao18@alumni.imperial.ac.uk

Dear Dr Lukas Gudmundsson,
Dear Referees,

Thank you for your review and response. We are happy to see that overall both referees are satisfied with the manuscript. We would like to thank reviewer 1 for raising a lot of technical problems. We found that the citation files downloaded from Google Scholar were not properly read in EndNote, which caused wrong citations. In addition, he pointed out problems related to grammar, data availability and paper structure. These are all good suggestions, and we have revised the article according to his suggestions.

**Major changes and additions to the revised manuscript (please list):**
1. The manuscript has been proofread and rendered.
2. The article has been revised based on the comments of reviewer 1.

**Specific Responses:**

**Response to Reviewer #1:**

**Comment 1:** *Also, in the revised version the authors state that the meteorological dataset "SURF_CLI_CHN_MUL_DAY" would be freely available for global researchers. However, in the source code repository they write that "SURF_CLI_CHN_MUL_DAY has just been closed for sharing." This has to be updated in the manuscript as well, if possible with some further explanation as the given link leads to a 404 error (page not found) within a chinese web portal.*

**Reply 1:** This has been updated in the article. Regarding the reason for closing, the response we got is "policy adjustment", and it may be reopened in the future.

Line 451: "Upon submission, due to policy adjustments, the SURF_CLI_CHN_MUL_DAY dataset has just been closed for sharing ..."

The dead urls are also removed.

**Comment 2:** *The manuscript needs proof reading; there are a number of grammatical and spelling mistakes. For example: ...*

**Reply 2:** The manuscript has been proofread and rendered. Please see the track up file.

**Comment 3:** *formatting should be improved. For example: ...*

**Reply 3:** All the places mentioned have been revised as suggested.

*Comment 4:* Please check carefully all references for completeness and exactness

**Reply 4:** We found that the citation file downloaded from Google was not properly read in EndNote, which caused wrong citations. This has been resolved by manual modification in EndNote.

***Comment 5****: Structure I propose to move table 3 to the appendix*

**Reply 5:** Although other CAMELS data sets put this table in the text, we think this is a good suggestion, so we followed it.

Sincerely,

Zhen Hao